# Ca²⁺/CaM binding to CaMKI promotes IMA-3 importin binding and nuclear translocation in sensory neurons to control behavioral adaptation

Domenica Ippolito, Saurabh Thapliyal, Dominique A Glauser*

Department of Biology, University of Fribourg, Fribourg, Switzerland

**Abstract** Sensory and behavioral plasticity are essential for animals to thrive in changing environments. As key effectors of intracellular calcium signaling, Ca²⁺/calmodulin-dependent protein kinases (CaMKs) can bridge neural activation with the many regulatory processes needed to orchestrate sensory adaptation, including by relaying signals to the nucleus. Here, we elucidate the molecular mechanism controlling the cell activation-dependent nuclear translocation of CMK-1, the *Caenorhabditis elegans* ortholog of mammalian CaMKI/IV, in thermosensory neurons in vivo. We show that an intracellular Ca²⁺ concentration elevation is necessary and sufficient to favor CMK-1 nuclear import. The binding of Ca²⁺/CaM to CMK-1 increases its affinity for IMA-3 importin, causing a redistribution with a relatively slow kinetics, matching the timescale of sensory adaptation. Furthermore, we show that this mechanism enables the encoding of opposite nuclear signals in neuron types with opposite calcium-responses and that it is essential for experience-dependent behavioral plasticity and gene transcription control in vivo. Since CaMKI/IV are conserved regulators of adaptable behaviors, similar mechanisms could exist in other organisms and for other sensory modalities.

## Editor's evaluation

This work elucidates the molecular mechanism of CaMKI shuttling between nucleus and cytoplasm and its function in thermal memory and thermal avoidance behavior in *C. elegans*. The authors thereby establish a direct link between the state of a signal transduction pathway, neuronal activity, and a complex behavioral output.

*For correspondence: dominique.glauser@unifr.ch

**Competing interest:** The authors declare that no competing interests exist.

## Introduction

Behavioral adaptation is essential for animal survival in a continuously changing natural environment. Any type of action that animals actuate as protection from external conditions, potentially harmful for their welfare, are part of behavioral adaptation strategies. These defensive behaviors primarily rely on specific sensory neurons, called nociceptors, which detect noxious stimuli and relay the information downstream in the nervous system to direct animal behavior (*Woolf and Ma, 2007*). Not every stimulus is able to activate nociceptive pathways, but only those with an intensity overcoming a certain activation threshold, which is not fixed and can be modulated under different circumstances. The presence of other type of stress (*Amit and Galina, 1986*), as well as previous, prolonged or repeated exposures to a noxious stimulus (*Schild et al., 2014*), is able to shift the threshold up, inducing a desensitization, or down, inducing a sensitization. In humans, the development of some health problems, like chronic pain, can be related to maladaptive sensory plasticity (*Pace et al., 2006*; *Gold and Gebhart, 2010*). Therefore, understanding the molecular mechanisms implicated in the modulation

of sensory plasticity, in particular in nociceptive pathways, could lead to potential therapeutic applications in the long term. Mammalian research models display some limitations, notably with respect to the size of the nervous system, the complexity and heterogeneity in nociceptor neuron populations, and animal suffering-related ethical concerns, which are particularly salient for studies on nociception and pain (*Gold and Gebhart, 2010*; *Dubin and Patapoutian, 2010*). As a complementary model, the roundworm *Caenorhabditis elegans* is a powerful and tractable genetic model, which exhibits considerable sensory plasticity. Worms are able to remember many types of environmental stimuli and display remarkable associative learning capabilities (*Ardiel and Rankin, 2010*). Avoidance behaviors, which *C. elegans* executes in response to noxious stimuli, constitute useful paradigms to study nociceptive plasticity (*Schild et al., 2014*; *de Bono and Maricq, 2005*; *Meisel and Kim, 2014*; *Komuniecki et al., 2012*).

*C. elegans* engages multiple thermosensory behaviors in order to stay away or escape from noxious heat, when placed in different regions of a spatial thermogradient reaching harmful temperatures (>26°C for the worm) (*Schild and Glauser, 2013*; *Glauser, 2013*). When acutely stimulated with heat, worms produce a robust, stereotyped thermal avoidance behavior, consisting in a worm reversal with a short backward movement, followed by a reorientation maneuver (*Wittenburg and Baumeister, 1999*; *Ghosh et al., 2012*). This withdrawal behavior relies on two pairs of head primary thermosensory neurons named AFDs and FLPs (*Liu et al., 2012*). AFD neuron also plays a major role in controlling thermotaxis in the innocuous temperature range, being both an exquisitely sensitive thermal detector and an essential memory and plasticity locus (*Goodman and Sengupta, 2018*). Indeed, AFD can detect tiny changes in temperature (*Ramot et al., 2008b*) and produce intracellular calcium transients in response to temperature elevations above a certain threshold, which depends on past thermosensory experience and which is determined by cell-autonomous mechanisms (*Kobayashi et al., 2016*). FLP neurons are thermosensors responding over a broad thermal range, from 8 to 38°C, with a steeper sensitivity in the noxious heat range (*Saro et al., 2020*). Temperature is encoded in FLP via a tonic signaling mode, with the activity level and steady-state intracellular calcium concentration reflecting the currently experienced temperature (*Saro et al., 2020*). The primary FLP thermosensory response to short-lasting heat stimuli seems largely nonadaptable, remaining unaffected by past thermal history. However, many lines of evidence indicate that FLP outputs can be modulated (*Schild et al., 2014*; *Saro et al., 2020*), taking part in the experience-dependent plasticity of the nociceptive response. Indeed, upon prolonged or repeated exposure to noxious heat, worms increase their thermal threshold for heat responsiveness through a desensitization, analgesia-like effect (*Schild et al., 2014*; *Lia and Glauser, 2020*).

In both AFD and FLP, as well as additional sensory neurons, experience-dependent plasticity involves the cell-autonomous activity of the calcium/calmodulin (CaM)-dependent protein kinase-1 (CMK-1) (*Schild et al., 2014*; *Satterlee et al., 2004*; *Yu et al., 2014*; *Neal et al., 2015*; *Moss et al., 2016*). CMK-1 is a Ser/Thr protein kinase widely expressed throughout the nervous system. It is the only homologue of the mammalian CaM kinases I and IV (CaMKI and CaMKIV) both functioning as intracellular $Ca^{2+}$ signaling effector in the nervous system (*Soderling, 1999*). CaM kinases play a role in processes including gene transcription, signal transduction, synaptic development and plasticity, memory and experience-dependent behaviors (*Satterlee et al., 2004*; *Hook and Means, 2001*; *Wayman et al., 2008b*; *Dahiya et al., 2019*; *Ardiel et al., 2018*). CMK-1 signaling pathway is functionally conserved from nematodes to humans (*Eto et al., 1999*). CMK-1 protein topology is also well-conserved. It comprises an extended N-terminal kinase catalytic domain and a C-terminal regulatory domain, which includes an autoinhibitory domain that partially overlaps with a CaM binding domain (*Hook and Means, 2001*; *Goldberg et al., 1996*). In the absence of $Ca^{2+}$/CaM binding, the regulatory domain maintains the catalytic domain in an inactive state. The binding of $Ca^{2+}$/CaM causes a conformational change in the protein that releases the autoinhibition. The subcellular localization of CaM kinases is subject to dynamic regulation. Hence, the nuclear localization of δCaMKI, γCaMKI, and some CaMKII isoforms depends on the cell-activation state (*Sakagami et al., 2005*; *Cohen et al., 2016*; *Schulman, 2004*). Whereas the mechanisms causing the translocation of CaMKII in a cell-activation dependent manner are well-characterized (*Zalcman et al., 2018*), those controlling CaMKI nuclear translocation and their physiological relevance in vivo are less well understood.

In FLP, CMK-1 plays a dual role in the heat-avoidance behavior. At innocuous temperatures, CMK-1 is localized principally in the cytoplasm, where it contributes to maintain a low behavioral response

threshold to promote heat avoidance. Within 1 hr of acclimation at 28°C (moderately noxious temperature), CMK-1 progressively translocates into the nucleus, where it produces the opposite effect, raising the noxious heat threshold to reduce avoidance (*Schild et al., 2014*). In AFD, CMK-1 also changes its subcellular localization upon temperature shifts, during which cultivation temperature memory is reset (*Yu et al., 2014*). The subcellular redistribution of CMK-1 thus appears to have a major role in adjusting neuron outputs according to past sensory experience. However, the molecular mechanisms through which sensory experience controls CMK-1 subcellular localization are largely unknown.

Here, we show how long-term cell-autonomous calcium activity is integrated to regulate CMK-1 localization in FLP and AFD thermosensory neurons and demonstrate the functional importance of this translocation mechanism in vivo. The proposed mechanism involves the dynamic nucleocytoplasmic shuttling of CMK-1 via an active transport, based on functional nuclear export sequence (NES) and nuclear localization signal (NLS) and their differential recruitment according to intracellular calcium levels. In particular, high intracellular calcium and $Ca^{2+}$/CaM binding to CMK-1 unmask an NLS localized in the N-terminal lobe of CMK-1 and increase the affinity for IMA-3 importin to favor progressive nuclear accumulation. Interestingly, long-term growth temperature has an opposite impact on CMK-1 localization in FLP as compared to AFD, in which CMK-1 is excluded from the nucleus at high temperatures (25°C). This opposite CMK-1 behavior is linked to an inverted long-term impact of temperature on intracellular calcium concentration in the two neuron types. We furthermore substantiate the physiological relevance of this mechanism in genome-edited lines with altered NLS and NES sequences, showing that they are essential for proper behavioral adaptation and gene expression.

## Results
### CMK-1 translocates in the nucleus of FLP upon prolonged exposure to heat

In order to evaluate how CMK-1 subcellular localization is regulated prior to and during adaptation, we focused on the FLP thermonociceptor neurons, in which a robust change in localization is easily monitored with a CMK-1::mNeonGreen (mNG) protein fusion (*Schild et al., 2014*; *Hostettler et al., 2017*). In naïve animals (grown at 20°C), CMK-1 is mostly localized in the cytoplasm of FLP (*Figure 1B*). Exposure to moderately noxious temperature (28°C) causes CMK-1 to progressively translocate into the nucleus, being slightly enriched in the nucleus after 90 min (*Figure 1B and C*). These observations recapitulate previous findings obtained with different fluorescent tags (*Schild et al., 2014*; *Hostettler et al., 2017*).

### CMK-1 cytoplasmic accumulation in unstimulated FLP neurons depends on an active transport and an intrinsic nuclear export sequence

Since CMK-1 subcellular localization varies according to cell types and conditions, and since putative NLS and NES are predicted from its primary sequence (*Schild et al., 2014*), CMK-1 localization may result from a dynamic equilibrium between nuclear import and export drives via canonical export/import pathways (*Sorokin et al., 2007*). To test this general working model, we started by addressing the mechanisms that promote the cytosolic accumulation of CMK-1 in FLP at 20°C. Two candidates NES were identified thanks to the NESSential prediction tool (*Fu et al., 2013*): NES[288-294] and NES[315-323] (superscript indexes indicate the residue positions, *Figure 1A*). We expressed CMK-1 mutant proteins affecting each of these NES candidates and examined the impact on CMK-1 localization in FLP at 20°C. Disrupting the NES[288-294] candidate sequence with a V292A/V294A double mutation or a Δ288–294 deletion prevented the cytoplasmic expression of CMK-1 and even yielded a nuclear accumulation (*Figure 1D*). In contrast, NES[315-323] candidate mutants displayed a cytoplasmic enrichment similar to that of wild-type CMK-1. These included I315A single mutant, V321A/V323A double mutant, I315A/V321A/V323A triple mutant, as well as Δ318–324 and Δ315–323 mutants, in which the NES candidate region is deleted in part or in full (*Figure 1D*). Collectively, these data indicate that the NES[288-294], but not the NES[314-323], is a key determinant of CMK-1 cytoplasmic localization in FLP at 20°C.

Because NES[288-294] sequence matches the binding consensus for exportins, we further tested whether CMK-1 is actively transported out of the nucleus at 20°C in FLP via this canonical pathway. Inhibiting exportins with the leptomycin B pharmacological inhibitor (*Kudo et al., 1999*) caused CMK-1::mNG nuclear translocation in FLP neurons at 20°C (*Figure 1E*).

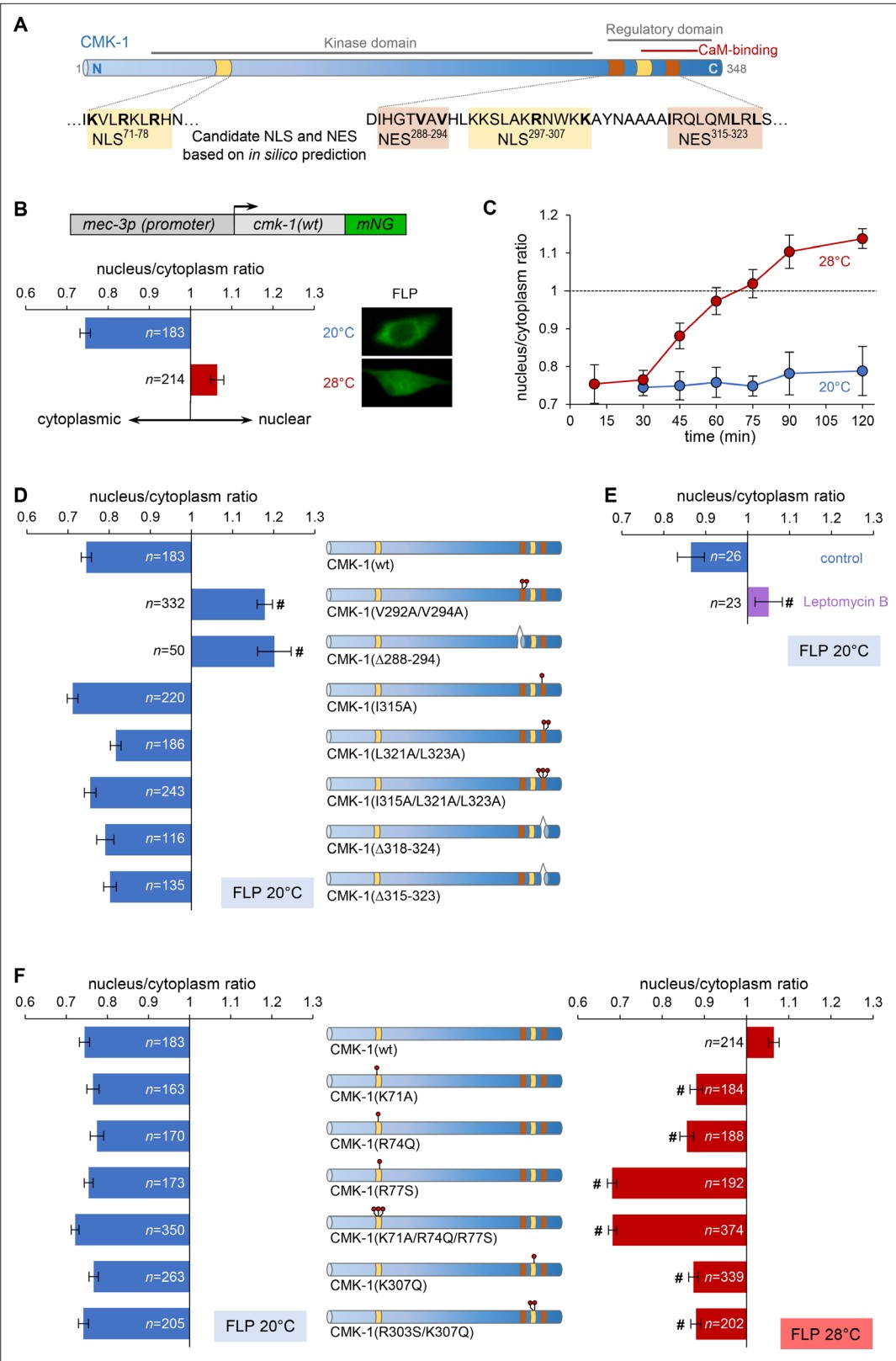

**Figure 1.** Specific nuclear export sequence (NES) and nuclear localization signal (NLS) control CMK-1 localization in response to temperature in FLP. (**A**) Schematic of CMK-1 topology highlighting the localization of in silico-predicted NES and NLS and their sequence. (**B**) Subcellular localization of CMK-1(wt)::mNeonGreen (mNG) reporter expressed in FLP via the depicted transgene (top). Average nuclear/cytoplasm fluorescent signal ratio

*Figure 1 continued on next page*

*Figure 1 continued*

(± SEM, left) and representative confocal micrographs (right) showing heat-evoked nuclear translocation of wild-type CMK-1 in young adult FLP neurons after 90 min at 28°C as compared to control at 20°C. (**C**) Kinetics of CMK-1::mNG nuclear accumulation. Data as nuclear/cytoplasmic fluorescent signal ratio average (± SEM). (**D, E**) Subcellular localization of CMK-1::mNG reporters carrying the depicted mutations in candidate NES (**D**), as well as following 90 min incubation with 50 µM leptomycin B or vehicle control (**E**). Data as nuclear/cytoplasmic signal ratio average (± SEM). (**F**) Same as for panel (**D**), but with mutations in the depicted candidate NLS in animals incubated 90 min at 20°C (blue, left panel) or 90 min at 28°C (red, right panel). #p<0.001 versus CMK-1(wt) by Bonferroni contrasts. The number of animals scored in each condition is indicated in the figure (n). Experiments reported in panels (**B**), (**D**), and (**F**) were run in parallel and the CMK-1(wt) dataset is common across these panels.

The online version of this article includes the following source data and figure supplement(s) for figure 1:

**Source data 1.** Summary statistics and raw data for *Figure 1*.

**Figure supplement 1.** CMK-1 localization data distribution in mutants for nuclear export sequence (NES) and nuclear localization signal (NLS) candidates.

**Figure supplement 2.** CMK-1 localization data distribution upon leptomycin B treatment.

**Figure supplement 3.** Illustration of regions of interest (ROI) definition.

Collectively, these results suggest that the predominant cytoplasmic expression of CMK-1 at 20°C in FLP requires NES[288-294] and relies on an exportin-dependent active nuclear export mechanism.

## Specific nuclear localization sequences on CMK-1 are required for nuclear translocation

Our next goal was to clarify the mechanisms causing CMK-1 to accumulate in the nucleus upon exposure to moderately noxious temperatures (28°C, 90 min). Hypothesizing that the importin pathway could be involved, we started by examining the role of two candidate NLS, defined with an in silico prediction tool (*Nguyen Ba et al., 2009*): NLS[71-78] and NLS[297-308]. Mutations affecting the NLS[71-78] candidate region all significantly impaired the nuclear relocalization of CMK-1 after prolonged heat exposure (*Figure 1F*, right). The inhibitory effect was partial in CMK-1(K71A) and CMK-1(R74Q) single mutants and complete in the CMK-1(R77S) single mutant, as well as in the CMK-1(K71A/R74Q/R77S) triple mutant. Introducing mutations in the NLS[297-307] candidate region (K307Q single mutant or a R302S/K307Q double mutant) also produced a partial reduction in CMK-1 nuclear accumulation. In contrast, none of the NLS-affecting mutations modulated the baseline cytoplasmic expression at 20°C (*Figure 1F*, left), suggesting that these elements might not be active in the absence of prolonged cell stimulation.

Taken together, our results suggest that (i) the canonical importin pathway may be involved, (ii) NLS[71-78] and NLS[297-308] are only engaged upon prolonged thermal stimulation, and (iii) NLS[71-78] plays a major role and NLS[297-308] a more minor role in driving CMK-1 nuclear accumulation.

## Intracellular calcium controls CMK-1 subcellular localization in FLP

Since FLP activity is controlled by temperature (*Saro et al., 2020*), we hypothesized that the temperature-evoked CMK-1 relocalization could reflect the prolonged cell-autonomous activity of FLP, independently from the inputs of additional neurons. Consistent with this model, we found that a prolonged heat treatment could still cause CMK-1 nuclear relocalization in *unc-13* mutants with impaired synaptic transmission (*Richmond et al., 1999*) and in *unc-31* mutants with impaired neuropeptide signaling (*Speese et al., 2007*; *Figure 2A*). These results suggest that the mechanisms leading to temperature-dependent CMK-1 nuclear relocalization could primarily depend on cell-autonomous heat-evoked FLP activity. However, we cannot rule out a redundant function of *unc-13* and *unc-31* gene products, nor the implication of electrical synapses.

Long-lasting thermal changes are mirrored by fluctuations in steady-state intracellular Ca$^{2+}$ levels in FLP (*Saro et al., 2020*). In order to evaluate the impact of intracellular Ca$^{2+}$ on CMK-1 subcellular localization, we first tested if an intracellular Ca$^{2+}$ elevation is sufficient to trigger CMK-1 nuclear accumulation in the absence of heat stimuli. We used *unc-68(dom13)* gain-of-function mutants in which cytoplasmic Ca$^{2+}$ levels are chronically elevated in FLP due to the expression of a deregulated ryanodine receptor/UNC-68 (*Marques et al., 2019*). In comparison to wild-type, CMK-1 nuclear expression

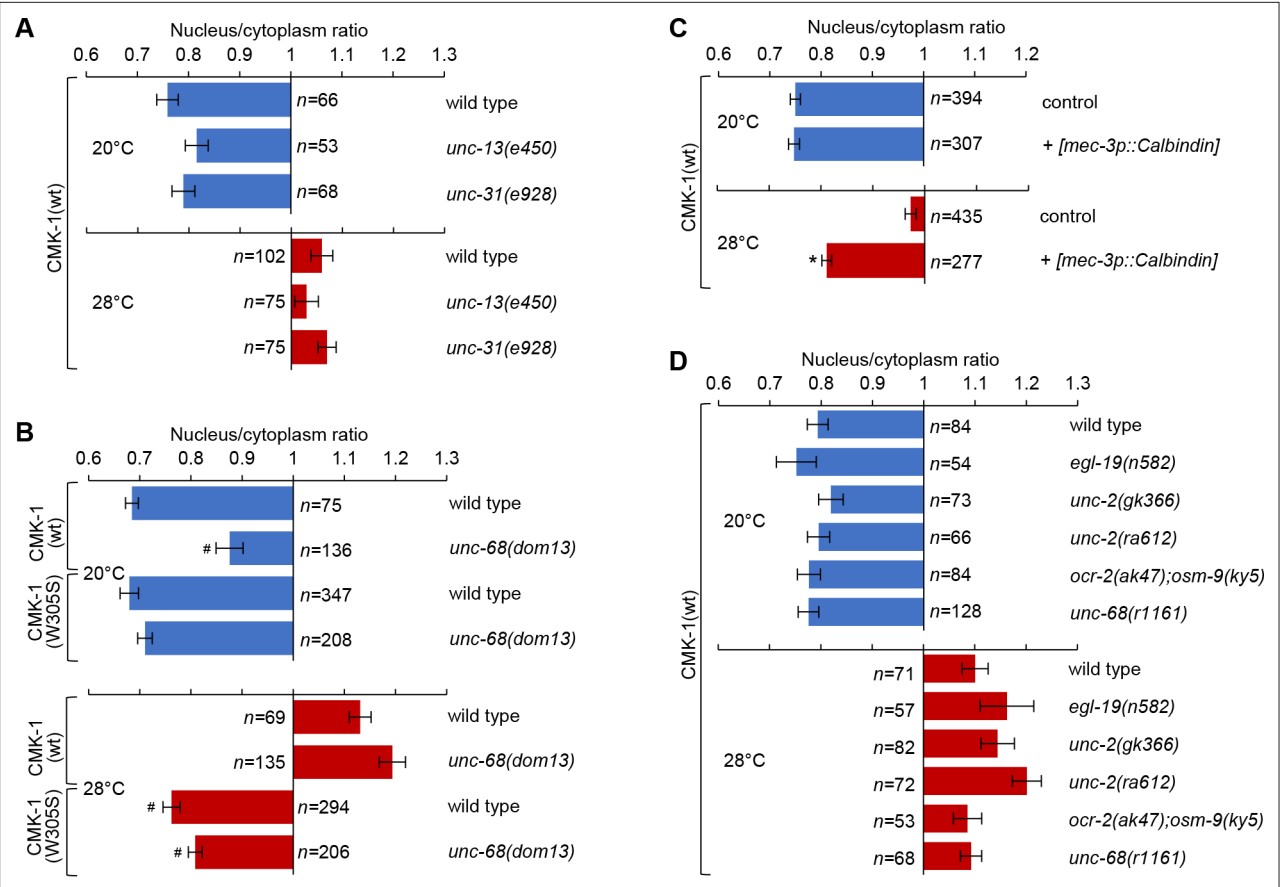

**Figure 2.** Intact cell-autonomous calcium signaling and CMK-1 Ca²⁺/CaM binding ability are essential to regulate CMK-1 subcellular localization in FLP. Subcellular localization of CMK-1(wt)::mNG reporters in FLP neurons of young adults of the indicated genotype, maintained 90 min at 20 or 28°C. Data as nuclear/cytoplasmic signal ratio average (± SEM). (**A**) Comparison showing no significant difference between wild-type and mutants affecting synaptic neurotransmission (*unc-13(e540)*) or dense core vesicle release (*unc-31(e928)*). (**B**) Subcellular localization of CMK-1(wt)::mNG and CMK-1(W305S)::mNG in wild-type and *unc-68(dom13)* mutant background. The *unc-68* gain-of-function mutation reduces the cytoplasmic accumulation of CMK-1(wt) at 20°C, but fails to do so when the CaM binding site is altered in CMK-1(W305S). n ≥ 135 animals. *p<0.001 versus wild-type; #p<0.001 versus CMK-1(wt) for the corresponding condition by Bonferroni contrasts. (**C**) Subcellular localization of CMK-1(wt)::mNG reporter in [*mec-3p::Calbindin*] transgenic animals, as well as non-transgenic animals coming from the same growth plates (control). The expression of the Calbindin calcium buffer inhibits CMK-1 nuclear translocation. n ≥ 277. *p<0.001 versus control at the same temperature by Bonferroni contrasts. (**D**) Same analysis as in (**A**), with indicated mutants affecting different calcium channels, but showing no statistically significant difference by Bonferroni contrasts.

The online version of this article includes the following source data and figure supplement(s) for figure 2:

**Source data 1.** Summary statistics and raw data for *Figure 2*.

**Figure supplement 1.** CMK-1 localization data distributions and statistical test results.

**Figure supplement 2.** FLP calcium concentration in Calbindin-expressing transgenic animals and calcium channel mutants.

**Figure supplement 3.** Parvalbumin expression in FLP reduces CMK-1 nuclear accumulation.

was stronger at 20°C in *unc-68(dom13)* mutants (*Figure 2B*, top). These data suggest that a chronic elevation in intracellular Ca²⁺ levels in the *unc-68(dom13)* mutants is sufficient to favor CMK-1 accumulation in the nucleus.

Next, we evaluated the impact of impairing the elevation of free Ca²⁺ intracellular concentration in transgenic animals overexpressing the Ca²⁺ buffer protein Calbindin (*Schumacher et al., 2012*). We generated a [*mec-3p::Calbindin*] transgene to express Calbindin in FLP and confirmed that it could reduce intracellular Ca²⁺ levels, as evidenced by a blunted heat-evoked YFP/CFP ratio increase in transgenic animals coexpressing the Ca²⁺ sensor cameleon YC2.3 (*Figure 2—figure supplement 2A*). Then, we evaluated CMK-1 localization and found that, whereas the localization of CMK-1 at 20°C was indistinguishable from that in wild-type, Calbindin overexpression in FLP significantly impaired its

accumulation in the nucleus at 28°C (*Figure 2C*). Overexpressing another $Ca^{2+}$ buffer, Parvalbumin, also significantly reduced the nuclear accumulation of CMK-1 at 28°C (*Figure 2—figure supplement 3*). In contrast, a mutant Parvalbumin with mutations affecting all its calcium binding sites did not impair CMK-1 localization (*Figure 2—figure supplement 3*), indicating that the effect of Parvalbumin involves its $Ca^{2+}$ binding activity. Collectively, the results of $Ca^{2+}$ buffering experiments suggest that an elevation of intracellular $Ca^{2+}$ in FLP is required for CMK-1 nuclear translocation upon prolonged thermal stimulation.

In an attempt to identify specific $Ca^{2+}$-permeable channels involved, we evaluated the impact of reduction-of-function mutations in the genes coding for the L- and N-type voltage-gated calcium channels (VGCC; *egl-19(n582)*, *unc-2(gk355)*, *unc-2(ra612)*), for transient receptor potential channels (*ocr-2(ak47)*; *osm-9(ky5)* double mutants) and for the ryanodine receptor (*unc-68(r1161)*). These mutations had previously been shown to affect FLP $Ca^{2+}$ response to short stimuli lasting between 30 s and 5 min (*Saro et al., 2020*). Contrary to our expectations, none of these mutations caused a significant alteration of CMK-1 localization (*Figure 2D*). These observations prompted us to examine if these mutations could actually impair steady-state $Ca^{2+}$ concentrations over a longer time frame matching that of CMK-1 translocation kinetics. Interestingly, none of the mutations significantly affected the $Ca^{2+}$elevation caused by a 60 min incubation at 28°C (*Figure 2—figure supplement 2B*). These results suggest either that other channels/transporters are involved or that at least some of these channels work in a redundant manner in regulating $Ca^{2+}$ steady-state levels and temperature-dependent CMK-1 nuclear accumulation.

Finally, reasoning that the binding of $Ca^{2+}$/CaM to CMK-1 could be directly involved in the relocalization mechanism, we examined the impact of a CMK-1(W305S) mutation, affecting a key residue necessary for the initiation of the CaM binding to CaM Kinases (*Matsushita and Nairn, 1998*). This CMK-1 mutation had a similar effect to the buffering of calcium with exogenous $Ca^{2+}$buffers, preventing CMK-1 nuclear translocation into the nucleus upon prolonged heat stimulation (28°C, *Figure 2B*, bottom). Furthermore, the W305S mutation was also able to counteract the CMK-1 nuclear accumulation observed in *unc-68(dom13)* mutant with permanently elevated calcium levels (*Figure 2B*).

Collectively, these results are consistent with a model in which prolonged thermal stimulation causes prolonged cell-autonomous FLP thermosensory activity, long-lasting changes in intracellular calcium concentration and binding of $Ca^{2+}$/CaM on CMK-1 to promote CMK-1 nuclear translocation.

## $Ca^{2+}$/CaM binding promotes CMK-1 nuclear localization independently of the $NES^{288-294}$ element

Our next goal was to understand how the binding of $Ca^{2+}$/CaM on CMK-1 can favor CMK-1 nuclear accumulation. One possible mechanism would consist in the masking of the $NES^{288-294}$ element by the binding of $Ca^{2+}$/CaM. Indeed, the $NES^{288-294}$ element is adjacent to the CaM binding domain and components of the export machinery might potentially compete with CaM for CMK-1 binding (*Figure 3A*, *NES masking model*). If this model is true, one would expect that the $NES^{288-294}$ element would be required to prevent CMK-1 nuclear accumulation at 28°C when the CaM-binding is disrupted by the W305S mutation. We therefore examined the localization of CMK-1(V292A/V294A/W305S) mutant proteins, in which both CaM binding and the $NES^{288-294}$ element are disrupted. Contrary to the predication made according to the *NES masking model*, we found that the triple mutant protein expression was cytoplasmic at 28°C, indistinguishable from that of the single W305S mutant (*Figure 3B*). Therefore, a functional $NES^{288-294}$ element is dispensable for the cytosolic retention at 28°C of CaM bindingdisruption mutant CMK-1. These data suggest that the masking of the $NES^{288-294}$ element upon CaM binding to CMK-1(wt) is unlikely to be a major mechanism promoting CMK-1 nuclear accumulation at 28°C.

## $Ca^{2+}$/CaM binding to CMK-1 increases its affinity for IMA-3 importin

An alternative mechanism through which $Ca^{2+}$/CaM binding could favor CMK-1 nuclear accumulation would be to trigger a conformational change that unmasks a functional NLS element (*Figure 3C*). Binding of importins would thereby be favored under high calcium conditions at 28°C, but not at 20°C when calcium levels are lower. The effect of mutations in the CMK-1 $NLS^{71-78}$ element is consistent with this model as they selectively impact CMK-1 localization at 28°C, but not at 20°C (*Figure 1F*). As presented in the next paragraphs, we further confirmed this *NLS unmasking model* by testing

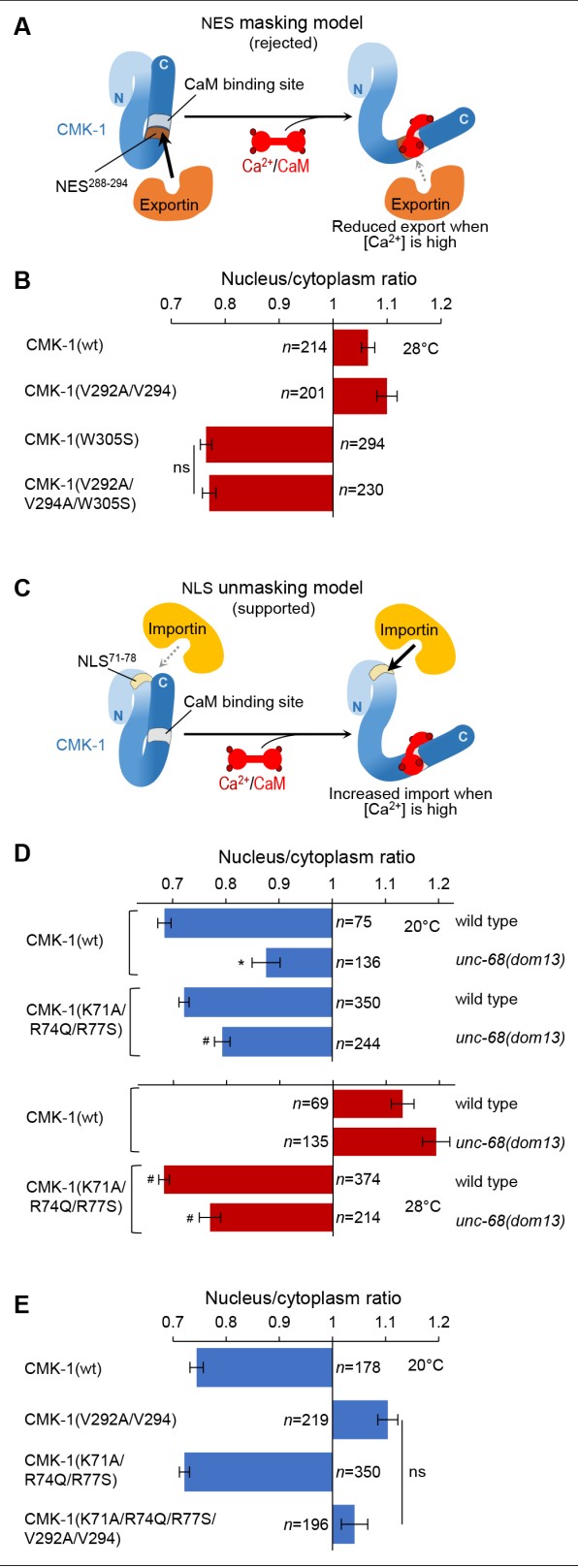

**Figure 3.** In vivo tests for nuclear export sequence (NES) masking and nuclear localization signal (NLS) unmasking models. (**A**) Schematic of the rejected model in which NES$^{288-294}$ would be masked by Ca$^{2+}$/CaM binding. (**B**) Subcellular localization of CMK-1::mNG reporters carrying the indicated mutations in the FLP neurons of animals exposed for 90 min at 28°C. Data as nuclear/cytoplasmic signal ratio average (± SEM), showing that the CaM-

*Figure 3 continued on next page*

*Figure 3 continued*

binding-disrupting mutation W305S prevents nuclear localization independently of the NES[288-294] element. (**C**) Schematic of the retained model in which NLS[71-78] would be unmasked by $Ca^{2+}$/CaM binding. (**D**) Subcellular localization of CMK-1(wt)::mNG and the NLS[71-78]-disrupting mutant CMK-1(K71A/R74Q/R77S)::mNG at 20 and 28°C in both wild-type and *unc-68(dom13)* backgrounds. Data as nuclear/cytoplasmic signal ratio average (± SEM), showing that the NLS[71-78]-disrupting mutations counteract the impact of the *unc-68* gain-of-function mutation. *p<0.001 versus wild-type; #p<0.001 versus CMK-1(wt) for the corresponding condition by Bonferroni post-hoc tests. Experiments were run in parallel to those reported in *Figure 2B*, and CMK-1(wt) data are common across the two figures. (**E**) Subcellular localization of CMK-1::mNG reporters carrying the indicated mutations disrupting either NES[288-294], NLS[71-78], or both of them. Data as nuclear/cytoplasmic signal ratio average (± SEM), showing no effect of NLS[71-78] disruption at 20°C, even when the NES[288-294] is impaired. ns, not significant.

The online version of this article includes the following source data and figure supplement(s) for figure 3:

**Source data 1.** Summary statistics and raw data for *Figure 3*.

**Figure supplement 1.** CMK-1 localization data distributions and statistical test results.

---

additional predictions regarding the localization of CMK-1 in vivo and directly quantifying protein affinities in vitro.

A first prediction of the *NLS unmasking* model is that the increased nuclear expression of CMK-1, triggered by *unc-68(dom13)* mutation (*Figure 3D*), will require a functional NLS[71-78] element. This was indeed the case as the expression of the NLS[71-78] mutant CMK-1(K71A/R74Q/R77S) was significantly more cytoplasmic than CMK-1(wt) in the *unc-68(dom13)*, high-calcium background at both 20 and 28°C (*Figure 3D*). We note that the nuclear accumulation is not fully blocked, which might suggest the existence of a NLS[71-78]-independent pathway. A second prediction of the model is that, because the NLS[71-78] element is masked at 20°C, the cytoplasmic exclusion observed in the NES[288-294] mutant at this temperature will not require the NLS[71-78] element. We therefore evaluated the localization of a CMK-1(K71A/R74Q/R77S/V292A/V294) mutant combining NLS[71-78]- and NES[288-294]-disrupting mutations (*Figure 3E*). According to our prediction, this mutant protein was excluded from the cytoplasm similarly to CMK-1(V292A/V294) mutant protein lacking only the functional NES[288-294] element. Therefore, at 20°C, when calcium levels are low and there is little CaM binding to CMK-1, the NLS[71-78] element is not engaged to regulate CMK-1 localization.

Next, we sought to more directly test the *NLS unmasking model* by quantifying the affinity between purified recombinant CMK-1 and importin-α in vitro. Indeed, importin-α are known to make direct interactions with NLS elements and CaM kinase IV was previously shown to interact with importin-α, even if the relevant NLS remained undefined in that case (*Kotera et al., 2005*). Three importin-α are encoded in the *C. elegans* genome (IMA-1, -2, and -3). Previous data suggest that IMA-3 is the only isoform expressed in somatic tissues (*Geles and Adam, 2001*). We confirmed the broad somatic expression of *ima-3* using an [*ima-3p::NLS::wrmScarlet*] reporter. Expressing cells included FLP neurons, as judged with the green/red co-labeling with the *mec-3p::CMK-1::mNG* reporter (*Figure 4A*). We therefore decided to conduct the in vitro protein-protein interaction analysis with IMA-3. HisTag::IMA-3, CMK-1(wt), and CMK-1(K71A/R74Q/R77S) were then produced in *Escherichia coli*, purified and protein-protein interactions quantified using the MicroScale Thermophoresis (MST) (*Jerabek-Willemsen et al., 2011*). Interaction plots are presented in *Figure 4B and C*, dissociation constants (Kd) summarized in *Table 1*, and a schematic interpretation is illustrated in *Figure 4D and E*. In control experiments, we could not detect any affinity between CaM and IMA-3, nor between CaM and CMK-1 in the absence of $Ca^{2+}$. In the absence of $Ca^{2+}$/CaM, CMK-1 had a relatively low but detectable affinity for IMA-3 (Kd ≈ 1.54 μM). In the presence of $Ca^{2+}$/CaM (2 mM $Ca^{2+}$ and 8 μM CaM), the affinity for IMA-3 raised by about approximately threefold (Kd ≈ 0.45 μM). No significant increase in affinity was seen with $Ca^{2+}$ only (2 mM $Ca^{2+}$, no CaM). These data suggest that the binding of $Ca^{2+}$/CaM to CMK-1 is sufficient to increase its affinity to IMA-3. This increased affinity effect was significantly reduced in CMK-1(K71A/R74Q/R77S) triple mutant, furthermore suggesting that the NLS[71-78] of CMK-1 mediates a major part of the $Ca^{2+}$/CaM-binding-dependent IMA-3 affinity increase.

Collectively, the results of in vivo protein localization analysis and in vitro protein affinity quantification are consistent with the *NLS[71-78] unmasking model* (schematically depicted in *Figure 3C*) to explain the increased nuclear accumulation in FLP neurons upon prolonged activation.

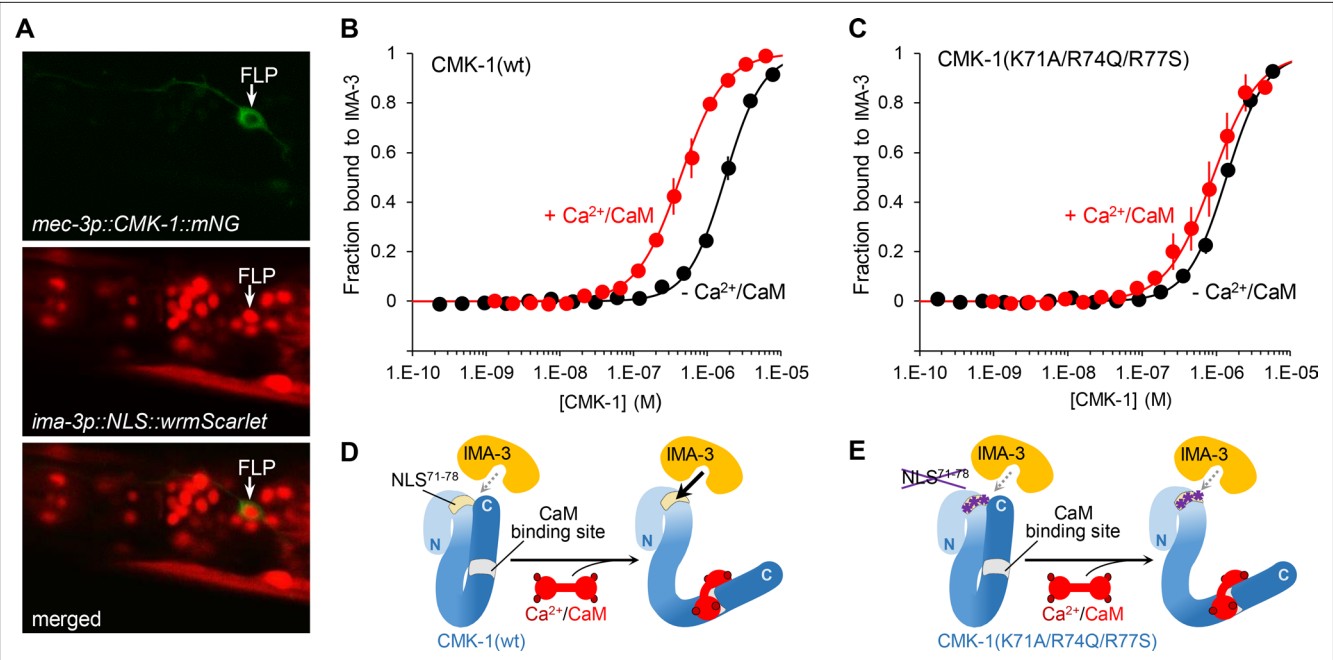

**Figure 4.** Ca$^{2+}$/CaM binding to CMK-1 enhances the affinity for IMA-3 importin-α in a NLS$^{71-78}$-dependent manner. (**A**) *ima-3* expression analysis with a transcriptional reporter. Representative confocal micrographs showing fluorescence signals in the head of an adult *[mec-3p::CMK-1::mNG; ima-3p::NLS::wrmScarlet]* transgenic animal. FLP cytoplasm is labeled in green (top). The nuclei of *ima-3*-expressing cells are labeled in red (middle). FLP is among the cells expressing *ima-3p* (bottom). (**B**) Binding curves for the CMK-1(wt)/IMA-3 interaction in the presence (red curve) or the absence (black curve) of Ca$^{2+}$/CaM, as determined by MicroScale Thermophoresis (see Materials and methods). Data as average of three replicates (± SEM). (**C**) Same analysis as in (**B**), but with CMK-1 (K71A/R74Q/R77S) mutant. Kd derived from fitting curves are reported in Table 1. (**D, E**) Schematic protein interaction models illustrating the increased affinity for IMA-3 when Ca$^{2+}$/CaM binds CMK-1(wt) (**D**) and the impact of NLS$^{71-78}$ mutations, which prevent CaM-dependent IMA-3 affinity increase.

The online version of this article includes the following source data for figure 4:

**Source data 1.** Summary statistics and raw data for *Figure 4*.

## CMK-1 NLS$^{71-78}$ and NES$^{288-294}$ elements are essential for thermal avoidance responsiveness and plasticity

In order to demonstrate the physiological relevance of the NLS$^{71-78}$ and NES$^{288-294}$-dependent control of CMK-1 subcellular localization, we engineered mutants with corresponding point mutations with CRISPR/Cas9 genome editing: *cmk-1(syb1435)*, encoding an R77S mutation that disrupts the NLS$^{71-78}$ element, and *cmk-1(syb1375)*, encoding for the V292A/V294A mutations that disrupt the NES$^{288-294}$ element (*Figure 5A*). While the NLS mutant appeared to be superficially wild-type as regards

**Table 1.** In vitro interaction between CMK-1 and IMA-3.

| IMA-3 | CMK-1 | CaM | Ca$^{2+}$ | Kd (mean ± SEM, μM) | $R^2$ fit | p vs. -CaM/ -Ca$^2$ control | p vs. CMK-1(wt) |
|---|---|---|---|---|---|---|---|
| + | wt | - | - | 1.54 ± 0.09 | 0.991 | - | - |
| + | wt | + | + | 0.45 ± 0.05 | 0.989 | <0 .0001 | - |
| + | - | + | + | No detectable interaction | - | - | - |
| + | wt | - | + | 1.14 ± 0.19 | 0.970 | ns | - |
| + | K71A/R74Q/R77S | - | - | 1.22 ± 0.09 | 0.988 | - | ns |
| + | K71A/R74Q/R77S | + | + | 1.00 ± 0.15 | 0.963 | ns | <0.05 |

ns = not significant.

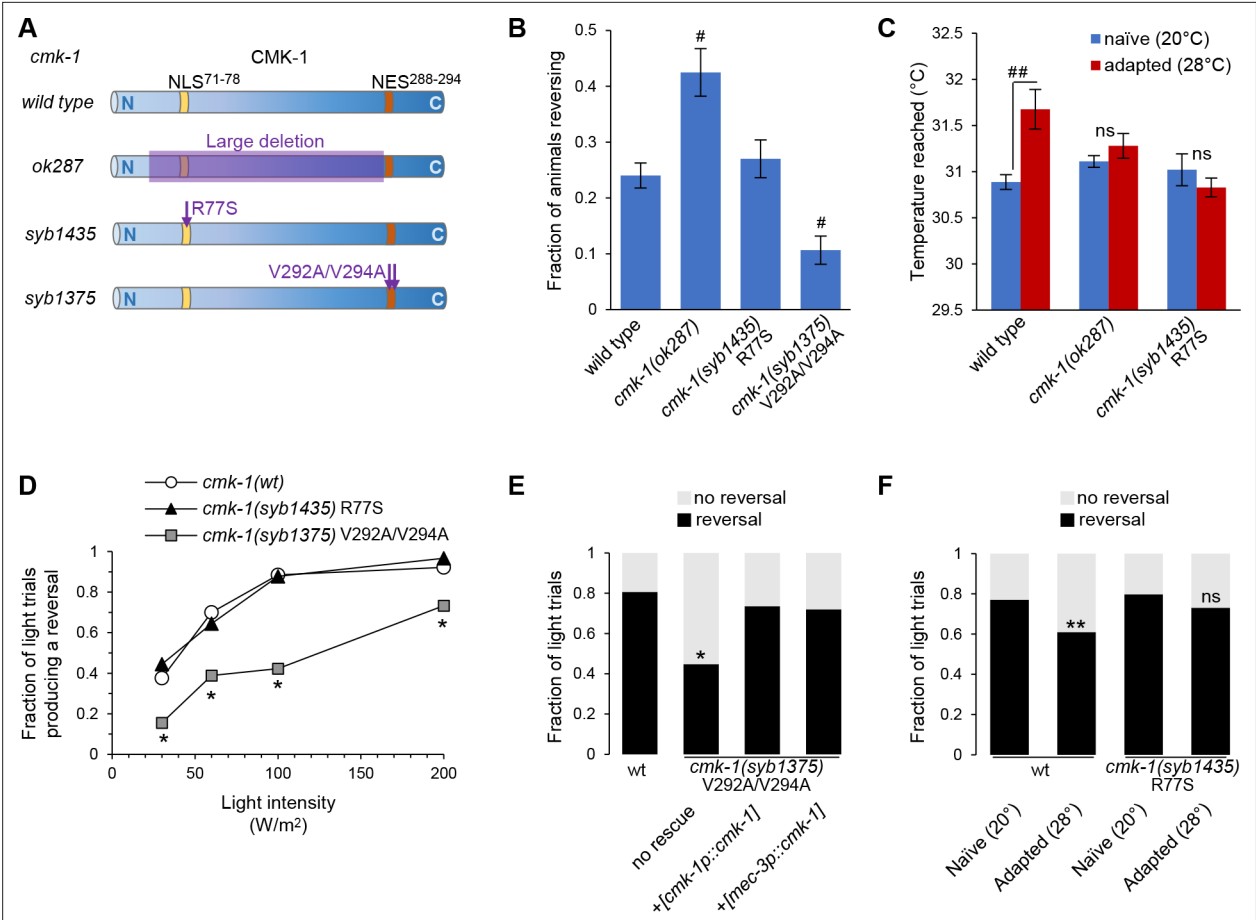

**Figure 5.** Engineered mutations affecting CMK-1 NLS$^{71-78}$ and NES$^{288-294}$ impact the adaptation of FLP-dependent behavior. (**A**) Schematic of CMK-1 protein highlighting the position of the deletion in *ok287* mutant and of the single-point mutations in *syb1435* and *syb1375* CRISPR/Cas9 engineered alleles affecting NLS$^{71-78}$ and NES$^{288-294}$, respectively. (**B**) Avoidance response to 4 s heat pulses (at 0.3–0.6 W /m$^2$, see Materials and methods) in adult animals of the indicated genotypes (average ± SEM, n ≥ 21 plates, each scoring at least 50 worms). #p<0.001 versus wild-type by Bonferroni contrasts. (**C**) Noxious heat thermogradient assays with animals of the indicated genotype maintained at 20°C (blue) or adapted for 1 hr at 28°C (red). Dispersal of animals reported as the temperature corresponding to the third quartile of the worm distribution in the spatial thermogradient. Vertical axis minimum was set to the lowest temperature in the thermogradient (29.5°C). Data as average (± SEM) of n ≥ 12 assays, each scoring more than 100 animals. ##p<0.001 by Bonferroni contrasts. ns, not significant. (**D**) FLP optogenetic analysis: light dose–response curves in young adult [*FLP::CoChR*] animals in response to 0.5 s light stimuli. Data as fraction of trials producing a response (n = 90 trials per genotype) showing a reduced response in *cmk-1(syb1375)* with constitutive nuclear CMK-1 expression. *p<0.001 versus *cmk-1(wt)* by Fisher's exact tests. (**E**) FLP optogenetic analysis: light avoidance rescue experiment in *cmk-1(syb1375)* mutant. Data acquired as in (**D**), using 100 W/m$^2$ light stimuli, and expressed as fraction of trials producing reversal or not. n ≥ 100 trials per genotype, each aggregating data from three independent rescue lines. *p<0.01 versus wild-type by Fisher's exact tests. Transgenes containing *cmk-1p* and *mec-3p* promoter both produced a rescue effect. (**F**) FLP optogenetic analysis: impact of heat adaptation. Data as in (**E**), in animals maintained at 20°C (naïve) or incubated for 90 min at 28°C (adapted), showing impaired adaptation in *cmk-1(syb1475)* mutants. n ≥ 120 light trials per condition. **p<0.01 versus naïve control by Fisher's exact test. ns, not significant.

The online version of this article includes the following source data and figure supplement(s) for figure 5:

**Source data 1.** Summary statistics and raw data for *Figure 5*.

**Figure supplement 1.** Optogenetic experiment controls.

morphology and gross locomotion phenotypes, the NES mutants had partially impaired locomotion with a tendency to coil, which prevented their characterization in some behavioral assays.

First, we evaluate heat-evoked reversals in wild-type and *cmk-1* mutants. Similar to previous findings (*Lia and Glauser, 2020*), loss of *cmk-1* in *cmk-1(ok287)* mutants increased the animal responsiveness to acute stimuli (*Figure 5B*). Whereas disrupting the NLS$^{71-78}$ in *cmk-1(syb1435)* R77S mutants had no impact on the heat-evoked response, disrupting the NES$^{288-294}$ in *cmk-1(syb1375)* V292A/

V294A double mutant significantly reduced animal responsiveness. In this latter case, the constitutive nuclear expression of the NES[288-294] mutant is thus linked to reduced aversive behaviors.

Second, we assessed heat-evoked avoidance using noxious heat thermogradient assays in which we scored how far animals spread in a noxious heat spatial thermogradient. We compared naïve animals (maintained at 20°C) and heat-adapted animals (maintained at 28°C for 1 hr prior to the assay). The adaptation treatment at 28°C caused a desensitization-like effect, shifting the distribution of wild-type animals toward higher temperatures (third quartile of their thermal distribution rising from 30.9 to 31.7°C, p>0.001, *Figure 5C*). Consistent with previous results (*Schild et al., 2014*), the loss of CMK-1 in *cmk-1(ok287)* mutants inhibited this adaptation effect (*Figure 5C*). We could not determine the impact of the NES[288-294]-affecting mutations because *cmk-1(syb1375)* mutants tended to coil and reverse more frequently, which limited their dispersal on the assay plate. The NLS[71-78]-affecting mutation in *cmk-1(syb1435)* R77S mutants did not affect the response of naïve animals (20°C), but impaired the desensitization effect upon adaptation at 28°C. These results highlight the importance of NLS[71-78]-dependent CMK-1 nuclear localization in the process mediating noxious heat desensitization.

## CMK-1 NLS[71-78] and NES[288-294] elements function to modulate FLP-dependent reversal behavior

Thermal avoidance behaviors in *C. elegans* are mediated by the FLP pathway, but also additional thermosensory pathways (*Liu et al., 2012*). In order to selectively address the role of CMK-1 NES[288-294] and NLS[71-78] in the FLP pathway, we analyzed FLP-specific optogenetic-evoked reversals. We previously showed that cell-autonomous CMK-1-dependent thermal avoidance plasticity in the FLP pathway was not linked to the modulation of FLP thermal sensitivity, but to the modulation of FLP neurotransmission (*Schild et al., 2014*). We therefore expected that CMK-1-dependent FLP output modulation would be detectable with an optogenetic activation, even if this stimulation method bypasses FLP thermosensation.

First, we assessed the impact of NES and NLS mutations in naïve animals grown at 20°C. As compared to reversals in wild-type, the reversals caused by FLP optogenetic activation were significantly reduced in *cmk-1(syb1375)* V292A/V294A double mutants with altered NES[288-294] (*Figure 5D*). In contrast, *cmk-1(syb1435)* R77S mutants with altered NLS[71-78] responded like wild-type. To determine if the NES mutation acts cell-autonomously in FLP, we conducted cell-specific rescue experiments with the *mec-3* promoter driving expression in FLP, but not in any of the downstream neurons mediating reversals. We observed a marked rescue effect, which was similar to that obtained with the endogenous *cmk-1* promoter (*Figure 5E*). Whereas we cannot exclude the implication of other neurons as additional sites of action for CMK-1, our data suggest that NES[288-294]-controlled CMK-1 cytoplasmic retention in FLP promotes reversals by upregulating FLP outputs.

Second, we analyzed the impact of thermal adaptation treatments on FLP-evoked reversals. A prolonged heat treatment (28°C for 90 min) reduced the reversal responses evoked by the optogenetic activation of FLP in wild-type animals (*Figure 5F*), confirming that heat-evoked plasticity entails a responsiveness reduction taking place downstream of FLP thermal sensitivity. This adaptation effect was blocked by the NLS[71-78]-affecting mutation in *cmk-1(syb1435)* mutants.

Collectively, these results indicate that the NLS[71-78] and NES[288-294]-dependent control of CMK-1 subcellular localization regulates FLP outputs and that NLS[71-78]-dependent nuclear accumulation is essential to dampen the FLP pathway response after persistent stimulations.

## Growth temperature impact on CMK-1 localization in AFD

We next sought to test if the CMK-1 localization control mechanism identified in FLP was also engaged in other neurons. We focused on the AFD thermosensory neurons, in which thermal shifts in the 15–25°C range were previously shown to affect CMK-1 localization (*Yu et al., 2014*). Animals were grown at 20°C and then shifted to 15 or 25°C for an overnight incubation. In animals incubated at 15°C, we observed that a CMK-1(wt)::mNG fusion was enriched in the nucleus of AFD (*Figure 6A and B*). In contrast, CMK-1 localization was cytoplasmic at 25°C (*Figure 6A and B*). Hence, the impact of long-term thermal history on CMK-1 localization in AFD is the opposite to that in FLP.

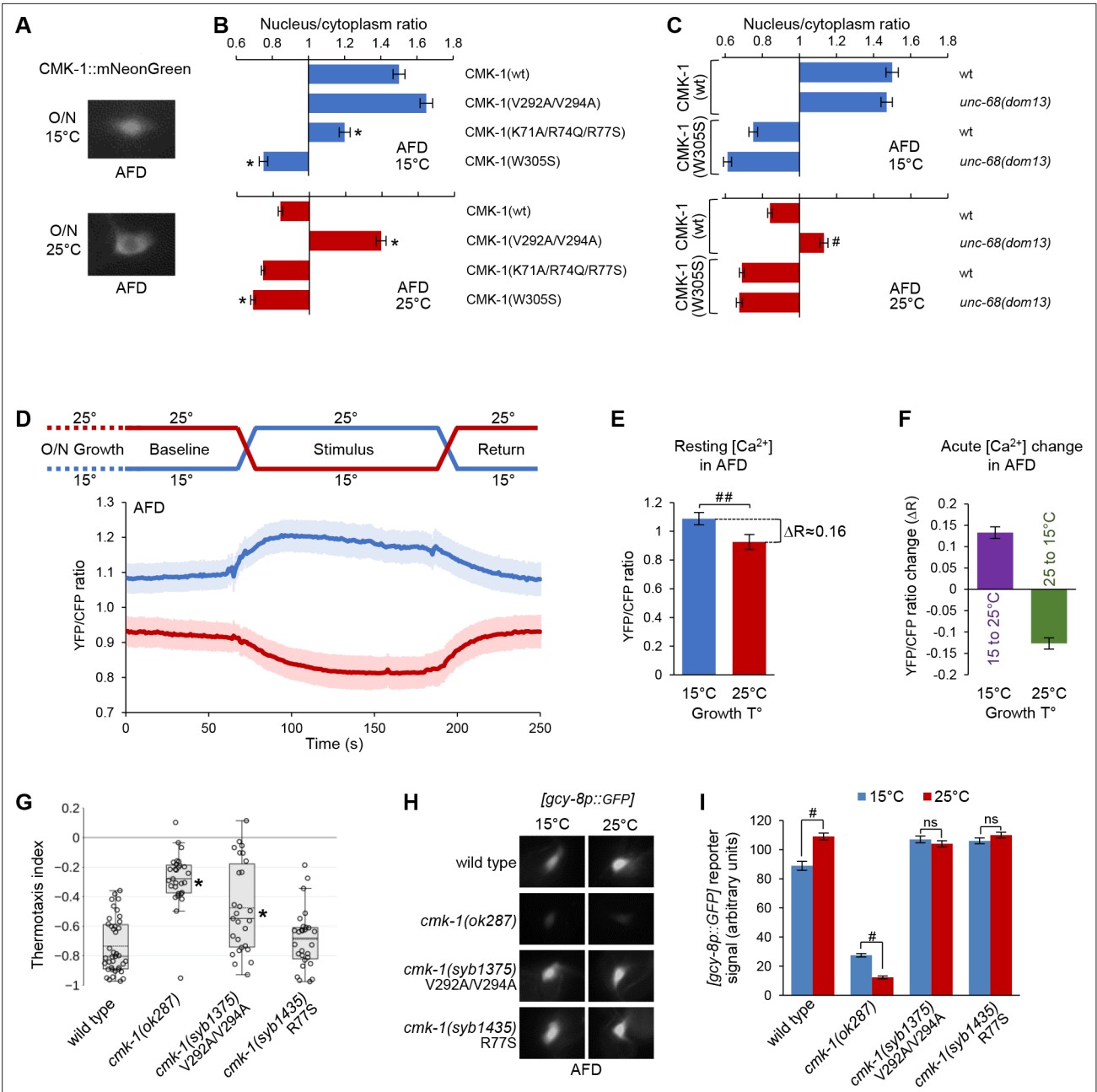

**Figure 6.** CMK-1 subcellular localization control and function in AFD. (**A**) Representative epifluorescence micrographs showing the subcellular expression pattern of CMK-1(wt)::mNG in AFD neurons of young adult animals maintained overnight at 15°C or at 25°C. (**B, C**) Subcellular localization of wild-type and indicated CMK-1::mNG mutants at 15°C (blue) and at 25°C (red) in wild-type and *unc-68(dom13)* background. Data as nuclear/cytoplasmic signal ratio average (± SEM) of n ≥ 125 animals. *p<0.01 versus CMK-1(wt); #p<0.01 versus same genotype at 20°C by Bonferroni contrasts. (**D**) Calcium levels in AFD cell bodies of [*ttx-1p:.YC2.3*] animals grown overnight (O/N) at 15 or 25°C and submitted to a 2 min temperature increase to 25°C or decrease to 15°C, respectively. Average calcium traces (blue and red lines) with SEM as lighter shades. n ≥ 16 animals. Note the impact of growth temperature on resting calcium levels (baseline and return periods). (**E**) Resting calcium-level quantification over the baseline period of data in panel (**D**), highlighting the impact of growth temperature (T°) (average ± SEM, n ≥ 16 animals). ##p<0.01 by Student's *t*-test. (**F**) Acute calcium-level changes caused by the indicated thermal up-steps or down-steps, quantified from data in panel (**D**) (average ± SEM, n ≥ 16 animals). (**G**) Thermotaxis assay result in young adult animals of the indicated genotypes grown at 20°C. n ≥ 25 assays, each scoring at least 80 animals. *p<0.01 versus wild-type by Bonferroni post-hoc tests. (**H**) Representative epifluorescence micrographs of young adult animals expressing a [*gcy-8p::GPF*] reporter in AFD neurons. Animals of the indicated genotypes were grown either at 15 or 25°C. n ≥ 80 animals per condition. (**I**) Quantification of GFP reporter signal from the data described in panel (**H**) (averages ± SEM). ##p<0.001 by Bonferroni contrasts. ns, not significant.

The online version of this article includes the following source data and figure supplement(s) for figure 6:

*Figure 6 continued on next page*

*Figure 6 continued*

**Source data 1.** Summary statistics and raw data for *Figure 6*.

**Figure supplement 1.** CMK-1 localization data distributions in AFD and statistical test results.

**Figure supplement 2.** Side-by-side comparison of FLP and AFD calcium response to temperature in the 15–25°C range.

**Figure supplement 3.** Expression of *ima-3* in AFD.

## Baseline intracellular calcium level in AFD inversely correlates with growth temperature

Could CMK-1 nuclear accumulation at 15°C reflect a long-term intracellular calcium elevation in AFD, like it does in FLP at high temperature? At first glance, this model looks counterintuitive because heat stimuli trigger intracellular calcium elevations in AFD (*Kimura et al., 2004*). However, AFD is fast-adapting and these elevations are transients, lasting at most for a few minutes when the stimulus is held constant (*Kimura et al., 2004*; *Clark et al., 2006*). Calcium levels over a longer time frame are more likely to be relevant for CMK-1 localization. We therefore compared AFD intracellular calcium levels between animals grown overnight at 15 and 25°C, respectively. We expressed the ratiometric YC2.3 cameleon sensor in AFD and compared the baseline YFP/CFP ratio across unstimulated animals. Remarkably, the baseline YFP/CFP ratio in worms grown and recorded at 15°C (mean = 1.09; sem = 0.03) was significantly higher than that measured in worms grown and recorded at 25°C (mean = 0.93; sem = 0.03; *Figure 6D*). The magnitude of the YFP/CFP ratio difference caused by differences in the overnight growth temperature ($\Delta R \approx 0.16$, *Figure 6E*) was of the same order of magnitude as those caused by acute thermal up-steps (15°C to 25°C, $\Delta R \approx 0.13$) or down-steps (25°C to 15°C, $\Delta R \approx 0.13$, *Figure 6E and F*). We conclude that baseline intracellular calcium level in AFD inversely correlates with growth temperature, a situation opposite to that in FLP (*Figure 6—figure supplement 2*).

## NLS[71-78] and NES[288-294] regulate CMK-1 subcellular localization in AFD neurons

Our findings about AFD indicate that an intracellular calcium elevation (at 15°C) is linked to an accumulation of CMK-1 in the nucleus, whereas an intracellular calcium reduction (at 25°C) is linked to a cytoplasmic expression. It seemed therefore plausible that CMK-1 subcellular localization could be controlled in AFD via the same calcium-dependent mechanism as in FLP. We made a series of observations confirming this hypothesis. First, the nuclear accumulation at 15°C was significantly impaired in the $Ca^{2+}$/CaM binding mutant CMK-1(W305S) and the NLS[71-78] mutant CMK-1(K71A/R74Q/R77S) (*Figure 6B*). Second, the UNC-68/ryanodine receptor *unc-68(dom13)* gain-of-function mutation caused a nuclear accumulation at 25°C (*Figure 6C*, bottom), suggesting that an elevation in calcium is sufficient to promote CMK-1 nuclear expression in AFD. Third, the nuclear accumulation in *unc-68(dom13)* mutant was completely abolished for CMK-1(W305S) mutant protein (*Figure 6C*, bottom), further pointing to the importance of $Ca^{2+}$/CaM binding. Fourth, disrupting the NES[288-294] in CMK-1(V292A/V294A) mutants leads to a constitutive nuclear expression in AFD irrespective of the temperature (*Figure 6B*). Finally, we also confirmed that *ima-3* is expressed in AFD (*Figure 6—figure supplement 3*). Collectively, these findings suggest that a similar mechanism control CMK-1 localization in FLP and AFD.

## CMK-1 subcellular localization mutations affect gene expression in AFD and thermotaxis

To further substantiate the physiological relevance of the NLS[71-78] and NES[288-294] elements in vivo in AFD, we evaluated the impact of engineered NLS and NES mutations on thermotaxis behavior and *gcy-8* gene expression, which are known to be regulated by CMK-1.

First, we tested the thermotactic response of animals grown at 20°C and deposited at 23°C in a linear temperature spatial gradient. Under these conditions, wild-type animals produced a robust negative thermotaxis response, moving toward cooler temperature (*Figure 6G*). Whereas *cmk-1(syb1435)* R77S mutants behaved like wild-type, the response was significantly reduced in *cmk-1(ok287)* loss-of-function and *cmk-1(syb1375)* V292A/V294A mutants. These data suggest that an intact NES[288-294] is essential for normal thermotaxis under our assay conditions.

Second, we addressed the impact of NLS and NES mutations on the expression of a transcriptional reporter for the AFD-specific gene *gcy-8*, whose transcription level was previously reported to rely on CMK-1 (*Satterlee et al., 2004*). In comparison to growth conditions at 15°C, *cmk-1(wt)* animals grown at 25°C had a stronger *gcy-8p::gfp* reporter activity (*Figure 6H and I*). The loss of CMK-1 in *cmk-1(ok287)* null mutants significantly reduced *gcy-8* transcription at either temperature and reverted the temperature impact. The growth-temperature effect was abolished in both *cmk-1(syb1435)* and *cmk-1(syb1375)* mutants, leaving expression levels closer to the situation in *cmk-1(wt)* animals grown at 25°C. The fact that both mutations caused the same effect despite having opposing effects on CMK-1 localization is in disagreement with a simple model where CMK-1 nuclear accumulation would suffice to regulate *gcy-8* transcription. Instead, the ability of CMK-1 to operate an active nucleocytoplasmic shuttling or to be localized in the right compartment at specific times might be essential for *gcy-8* gene transcription regulation.

Collectively, these results suggest that normal NES$^{288-294}$- and NLS$^{71-78}$-dependent CMK-1 subcellular localization control is essential to maintain the function of AFD.

## Discussion

Sensory and behavioral plasticity are essential for animals to thrive in changing environments. CaM kinases are ubiquitous signaling molecules able to convert intracellular calcium signals into specific phosphorylation patterns on target proteins and playing an important role in neural plasticity (*Soderling, 1999*; *Wayman et al., 2008a*). In particular, CMK-1 plays antagonistic roles in the cytoplasm and the nucleus of FLP thermonociceptors, contributing to adjust the output from these cells and to modulate avoidance behavior (*Schild et al., 2014*). While the importance of this subcellular compartment-specific signaling is well appreciated, little was known on how CMK-1 could operate localization switch according to recent sensory experience on a minute-to-hour timescale. Our work provides insight on the molecular factors governing CMK-1 subcellular localization in vivo and allows us to propose a new mechanism through which CMK-1 subcellular localization is coupled to cell activation and ensuing long-term calcium-level modulation. We furthermore show that this mechanism is not restricted to FLP neurons and is essential for proper behavioral plasticity and gene expression.

The proposed mechanism for controlled CMK-1 localization relies on three main features. First, the presence of two key localization signals on the CMK-1 protein: NLS$^{71-78}$ and NES$^{288-294}$, with sequences matching the consensus for recognition by importins and exportins, respectively. Considering our findings with exportin pharmacological inhibition and in vitro binding to the IMA-3 importin, these two sequence elements are likely to function via their respective canonical pathways. Second, the impact of Ca$^{2+}$/CaM-binding to CMK-1, which is sufficient to promote the interaction with IMA-3, via the NLS$^{71-78}$ sequence. Therefore, the binding of Ca$^{2+}$/CaM and ensuing conformational changes have a dual impact by (i) favoring CMK-1 catalytic activity and (ii) unmasking NLS$^{71-78}$ for nuclear import. Third, the long-term modulation of steady-state intracellular calcium levels in sensory neurons, which will produce relatively slow-developing effects on CMK-1 localization.

There are several precedents for activity-dependent nucleocytoplasmic translocation of proteins in neurons in mammals (*Schulman, 2004*; *Ch'ng et al., 2015*; *Crabtree and Olson, 2002*; *Chawla et al., 2003*). In these cases, the key mechanism is the phosphorylation status of the target protein, which is modulated in a cell activity-dependent manner by phosphatases (*Crabtree and Olson, 2002*) and kinases (*Schulman, 2004*; *Chawla et al., 2003*). Here, we reveal a novel mechanism for activity-dependent control of CaM kinase signaling, allowing to couple intracellular calcium elevation and Ca$^{2+}$/CaM binding to CMK-1 with its redistribution into the nucleus via the importin pathway. Based on in vitro analysis of permeabilized cells, *Sweitzer and Hanover, 1996* proposed that some protein cargos could be imported into the nucleus directly via a 'CaM-dependent nuclear protein import' mechanism, not requiring Ran-GTP, nor the canonical importin pathway. The exact mechanism through which CaM could promote cargo translocation in this pathway remains elusive. The mechanism that we propose here is different because the translocation of CMK-1 not only requires CaM binding, but also the presence of an intact NLS$^{71-78}$ sequence, which we showed to be a functional binding site for the IMA-3 importin-α. We consider that the most likely explanation for our data would be that the CaM-binding-induced conformational change,

which is well-documented in CaM kinase, could cause CMK-1 to expose the NLS[71-78] sequence in order to make it more accessible to the importin pathways, via a direct interaction with IMA-3.

In addition to the NLS[71-78]-dependent mechanism, several lines of evidence suggest the existence of one or more additional mechanisms able to promote CMK-1 nuclear entry or retention. Indeed, we noted that the NLS[71-78] mutation reduced but did not abolish the interaction with IMA-3 (*Figure 4C*). Furthermore, the NLS[71-78] mutation could not fully prevent the CMK-1 nuclear accumulation caused by the *unc-68(dom13)* mutation in FLP (*Figure 3D*) or by overnight cultivation at 15°C in AFD (*Figure 6B*). However, NLS[71-78]-independent pathways may not play a major role in FLP when calcium is elevated by a 90 min incubation at 28°C (full penetrance of the NLS mutation in FLP, *Figure 1F*). Further studies will be needed to delineate the exact nature of NLS[71-78]-independent mechanisms, and notably if they also involve IMA-3-dependent nuclear import and alternative NLS such as NLS[297-308].

The presence of functional NLS and NES suggests that CMK-1 localization results from a dynamic influx/efflux equilibrium. The kinetics of CMK-1 nuclear entry is rather slow, needing about 1 hr in FLP to abolish the cytoplasmic enrichment of the protein. Whereas our data do not exclude additional mechanisms, the threefold increase in IMA-3/CMK-1 affinity upon $Ca^{2+}$/CaM binding is likely to contribute to shift this equilibrium. A previous study addressing the localization of cargo proteins with systematic NLS variations showed that varying the affinity for importins was sufficient to modify the nuclear/cytoplasmic ratio of these cargos and defined the upper and lower *Kd* value limits for active import in yeast (*Hodel et al., 2001*). Based on this reference *Kd* scale, the CMK-1-/IMA-3 affinity would correspond to an undetectable import in the absence of CaM and to a moderately active import in the presence of CaM, which is in line with the relatively low CMK-1 nuclear enrichment observed, even when the NES is disrupted. Considering the relatively slow kinetics of the system, we speculate that it may actually serve to filter out the influence of local, short-lasting calcium peaks, in order for CMK-1 localization to reflect mostly the global changes occurring over longer timescales. This seems to be an appropriate strategy to modulate cellular outputs and gene expression during behavioral adaptations, which typically occur over longer timescales than calcium activity peaks in response to acute stimuli. The situation is particularly interesting in AFD, where acute calcium elevations caused by short thermal fluctuation could potentially be 'ignored,' to focus on the long-lasting, global impact that temperature has on resting calcium levels: in the case of AFD, a decrease in calcium favoring CMK-1 nuclear exclusion at higher temperatures.

AFD is an extensively studied thermosensory neuron (*Goodman and Sengupta, 2018*). To our knowledge, however, the long-term impact of temperature on resting intracellular calcium levels was not known. Previous calcium imaging studies considered shorter time frames and focused on relative calcium changes over a baseline, but did not address how this baseline was affected in different conditions (*Kobayashi et al., 2016*; *Kimura et al., 2004*; *Clark et al., 2006*; *Kimata et al., 2012*; *Hawk et al., 2018*; *Kuhara et al., 2011*; *Takeishi et al., 2016*; *Clark et al., 2007*). Here, by using a cameleon ratiometric indicator, we evaluated the impact of overnight temperature shifts and showed that high temperature (25°C) unexpectedly causes a decrease in resting calcium level as compared to animals at 15°C. This effect is the inverse of that seen in FLP (*Figure 6— figure supplement 2*). Therefore, AFD calcium responses include both the well-described short-term response to acute thermal change and the newly described resting calcium drift component, where temperature has the opposite impact and which serves as a major determinant for CMK-1 localization. We speculate that long-term drifts in resting calcium levels could play a role in the way the thermal response threshold is encoded and/or decoded to shape thermotactic plasticity. Additional studies will be required to better understand the role of AFD global calcium signaling and to identify the molecular machinery enabling AFD to produce both increase and decrease in calcium in response to temperature over different timescales.

In conclusion, we have identified a novel $Ca^{2+}$-dependent mechanism, which controls the intracellular locus of CaM kinase signaling, and which operates on longer time frames than the well-known kinase autoinhibitory control, in order to adjust gene expression and neuron function during adaptation. We speculate that this mechanism may be relevant for other sensory modalities, in particular for those neurons subject to long-term calcium changes, such as tonically signaling sensory neurons (*Busch et al., 2012*). Furthermore, it may be instrumental for plasticity mechanisms in other species due to the high conservation of $Ca^{2+}$ signaling throughout the evolution.

# Materials and methods

## Key resources table

| Reagent type (species) or resource | Designation | Source or reference | Identifiers | Additional information |
|---|---|---|---|---|
| Genetic reagent (*Caenorhabditis elegans*) | N2 | CGC; RRID:SCR_007341 | Wild-type | |
| Genetic reagent (*C. elegans*) | DAG439 | This study | *domSi439[mec-3p::cmk-1 (1–348)::mNG::unc-54 3'UTR] II* | Expression of CMK-1(wt)::mNG in FLP |
| Genetic reagent (*C. elegans*) | PHX1375 | This study | *cmk-1(syb1375) IV* | Allele encoding CMK-1 (V292A/V294A) Mutation made by genome editing (SunyBiotech, China) |
| Genetic reagent (*C. elegans*) | PHX1435 | This study | *cmk-1(syb1435) IV* | Allele encoding CMK-1(R77S) Mutation made by genome editing (SunyBiotech, China) |
| Genetic reagent (*C. elegans*) | DAG356 | *Marques et al., 2019* | *domIs272a[mec-3p::QF, mec-4p::QS, QUAS::CoChR, unc-122p::RFP]* | [FLP::CoChR] FLP optogenetic background, used as background for light-avoidance assays |
| Genetic reagent (*C. elegans*) | DAG821 | This study | *cmk-1(ok287) IV* | *cmk-1* loss-of-function allele 4× backcrossed |
| Genetic reagent (*C. elegans*) | DAG800 | *Schild et al., 2014* | *cmk-1(pg58) IV* | *cmk-1* gain-of-function allele 4× backcrossed |
| Genetic reagent (*C. elegans*) | DAG874 | *Marques et al., 2019* | *unc-68(dom13) V* | *unc-68* gain-of-function allele 4× backcrossed |
| Genetic reagent (*C. elegans*) | DAG927 | This study | *domSi439[mec-3p::cmk-1::mNG::3xFlag::unc-54 3'UTR] II; unc-31(e928) IV* | Expression of CMK-1(wt)::mNG in FLP in *unc-31* mutant background |
| Genetic reagent (*C. elegans*) | DAG928 | This study | *domSi439[mec-3p::cmk-1::mNG::3xFlag::unc-54 3'UTR] II; unc-13(e450) I* | Expression of CMK-1(wt)::mNG in FLP in *unc-13* mutant background |
| Genetic reagent (*C. elegans*) | DAG1032 | This study | *domSi439[mec-3p::cmk-1::mNG::3xFlag::unc-54 3'UTR] II;[unc-68(dom13)] V* | Expression of CMK-1(wt)::mNG in FLP in *unc-68* mutant background |
| Genetic reagent (*C. elegans*) | DAG1204 | This study | *cmk-1(ok287) IV; domIs272[mec-3p::QF, mec-4p::QS, QUAS::CoChR, unc-122p::RFP] II* | *cmk-1* loss-of-function allele in [FLP::CoChR] FLP optogenetic background for light-avoidance assays |
| Genetic reagent (*C. elegans*) | DAG1205 | This study | *cmk-1 (syb1375) IV; domIs272[mec-3p::QF, mec-4p::QS, QUAS::CoChR, unc-122p::RFP] II* | *cmk-1* genome-edited mutation coding for CMK-1(V292A/V294A) in [FLP::CoChR] FLP optogenetic background for light-avoidance assays |
| Genetic reagent (*C. elegans*) | DAG1206 | This study | *cmk-1(syb1435) IV; domIs272[mec-3p::QF, mec-4p::QS, QUAS::CoChR, unc-122p::RFP] II* | *cmk-1* genome-edited mutation coding for CMK-1(R77S) in [FLP::CoChR] FLP optogenetic background for light-avoidance assays |
| Genetic reagent (*C. elegans*) | GN2 | Gift frm Miriam Goodman | *oyIs17[gcy-8p::GFP]* | *gcy-8* transcriptional reporter |
| Genetic reagent (*C. elegans*) | VC220 | CGC; RRID:SCR_007341 | *cmk-1(ok287); oyIs17[gcy-8p::GFP]* | *gcy-8* transcriptional reporter in *cmk-1* loss-of-function background |
| Genetic reagent (*C. elegans*) | DAG1053 | This study | *cmk-1(syb1375) IV; oyIs17[gcy-8p::GFP]* | *gcy-8* transcriptional reporter in *cmk-1* genome-edited mutant background coding for CMK-1(V292A/V294A) |
| Genetic reagent (*C. elegans*) | DAG1054 | This study | *cmk-1(syb1435) IV; oyIs17[gcy-8p::GFP]* | *gcy-8* transcriptional reporter in *cmk-1* genome-edited mutant background coding for CMK-1(R77S) |
| Genetic reagent (*C. elegans*) | DAG977 | This study | *domSi439[mec-3p::cmk-1::mNG::3xFlag::unc-54 3'UTR] II; osm-9(ky10) ocr-2(ak47) IV* | Expression of CMK-1(wt)::mNG in FLP in *osm-9;ocr-2* double mutant background |

*Continued on next page*

*Continued*

| Reagent type (species) or resource | Designation | Source or reference | Identifiers | Additional information |
|---|---|---|---|---|
| Genetic reagent (*C. elegans*) | DAG978 | This study | *domSi439[mec-3p::cmk-1::mNG::3xFlag::unc-54 3'UTR] II; unc-68(r1161) V* | Expression of CMK-1(wt)::mNG in FLP in *unc-68* loss-of-function mutant background |
| Genetic reagent (*C. elegans*) | DAG979 | This study | *domSi439[mec-3p::cmk-1::mNG::3xFlag::unc-54 3'UTR] II; unc-2(gk366) X* | Expression of CMK-1(wt)::mNG in FLP in *unc-2* mutant background |
| Genetic reagent (*C. elegans*) | DAG980 | This study | *domSi439[mec-3p::cmk-1::mNG::3xFlag::unc-54 3'UTR] II; unc-2(ra612) X* | Expression of CMK-1(wt)::mNG in FLP in *unc-2* mutant background |
| Genetic reagent (*C. elegans*) | DAG981 | This study | *domSi439[mec-3p::cmk-1::mNG::3xFlag::unc-54 3'UTR] II; egl-19(n582) IV* | Expression of CMK-1(wt)::mNG in FLP in *egl-19* mutant background |
| Genetic reagent (*C. elegans*) | DAG565-566-567 | This study | *domEx565-566-567[mec-3p::cmk-1(Δ315–323)::mNG, unc-122p::RFP]* | Expression of CMK-1(Δ315–323)::mNG in FLP |
| Genetic reagent (*C. elegans*) | DAG576-577-578 | This study | *domEx576-577-578[mec-3p::cmk-1(Δ318–324)::mNG, unc-122p::RFP]* | Expression of CMK-1(Δ318–324)::mNG in FLP |
| Genetic reagent (*C. elegans*) | DAG592-593 | This study | *domEx592-593[mec-3p::CMK-1(Δ288–294)::mNG, unc-122p::RFP]* | Expression of CMK-1(Δ288–294)::mNG in FLP |
| Genetic reagent (*C. elegans*) | DAG703-704-705 | This study | *domEx703-704-705[mec-3p::cmk-1(V292A/V294A)::mNG, unc-122p::RFP]* | Expression of CMK-1(V292A/V294A)::mNG in FLP |
| Genetic reagent (*C. elegans*) | DAG706-707-708 | This study | *domEx706-707-708[mec-3p::cmk-1(K71A/R74A/R77S)::mNG, unc-122p::RFP]* | Expression of CMK-1(K71A/R74A/R77S)::mNG in FLP |
| Genetic reagent (*C. elegans*) | DAG727-728-729 | This study | *domEx727-728-729[mec-3p::cmk-1(L321A/L323A)::mNG, unc-122p::RFP]* | Expression of CMK-1 (L321A/L323A)::mNG in FLP |
| Genetic reagent (*C. elegans*) | DAG1009-1010-1011 | This study | *domEx1009-1010-1011[mec-3p::cmk-1(I315A)::mNG, unc-122p::RFP]* | Expression of CMK-1(I315A)::mNG in FLP |
| Genetic reagent (*C. elegans*) | DAG1012-1013-1014 | This study | *domEx1012-1013-1014[mec-3p::cmk-1(I315A/L321A/L323A)::mNG, unc-122p::RFP]* | Expression of CMK-1(I315A/L321A/L323A)::mNG in FLP |
| Genetic reagent (*C. elegans*) | DAG738-739-740 | This study | *domEx738-739-740[mec-3p::cmk-1(K307Q)::mNG, unc-122p::RFP]* | Expression of CMK-1(K307Q)::mNG in FLP |
| Genetic reagent (*C. elegans*) | DAG741-742-743 | This study | *domEx741-742-743[mec-3p::cmk-1(R302S/K307Q)::mNG, unc-122p::RFP]* | Expression of CMK-1(R302S/K307Q)::mNG in FLP |
| Genetic reagent (*C. elegans*) | DAG962-963-966 | This study | *domEx962-963-966[mec-3p::cmk-1(K71A)::mNG, unc-122p::RFP]* | Expression of CMK-1(K71A)::mNG in FLP |
| Genetic reagent (*C. elegans*) | DAG964-965 | This study | *domEx964-965[mec-3p::cmk-1(R74Q)::mNG, unc-122p::RFP]* | Expression of CMK-1(R74Q)::mNG in FLP |
| Genetic reagent (*C. elegans*) | DAG967-968-969 | This study | *domEx967-968-969[mec-3p::cmk-1(R77S)::mNG, unc-122p::RFP]* | Expression of CMK-1(R77S)::mNG in FLP |
| Genetic reagent (*C. elegans*) | DAG900-901-902 | This study | *domEx900-901-902[mec-3p::cmk-1(K71A/R74Q/R77S/V292A/V294A)::mNG, unc-122p::RFP]* | Expression of CMK-1(K71A/R74Q/R77S/V292A/V294A)::mNG in FLP |
| Genetic reagent (*C. elegans*) | DAG700-701-702 | This study | *domEx700-701-702[mec-3p::cmk-1(W305S)::mNG, unc-122p::RFP]* | Expression of CMK-1(W305S)::mNG in FLP |
| Genetic reagent (*C. elegans*) | DAG912-913-914 | This study | *domEx912-913-914[mec-3p::cmk-1(W305S/V292A/V294A)::mNG, unc-122p::RFP]* | Expression of CMK-1(W305S/V292A/V294A)::mNG in FLP |
| Genetic reagent (*C. elegans*) | DAG1029-1030-1031 | This study | *domEx1029-1030-1031[ima-3p::egl-13NLS::wrmScarlet::unc-54 3'UTR, unc-122p::GFP]; domSi439[mec-3p::cmk-1::mNG::3xFlag::unc-54 3'UTR] II* | Transcriptional reporter for *ima-3* driving the expression of a red nuclear marker in a background with FLP labeled in green |

*Continued on next page*

*Continued*

| Reagent type (species) or resource | Designation | Source or reference | Identifiers | Additional information |
|---|---|---|---|---|
| Genetic reagent (C. elegans) | DAG1244-1245-1246 | This study | *domEx1244-1245-1246[mec-3p::cmk-1(R77S)::mNG, unc-122p::RFP];[unc-68(dom13)]V* | Expression of CMK-1(R77S)::mNG in FLP in a *unc-68* gain-of-function background |
| Genetic reagent (C. elegans) | DAG1279-1280-1281 | This study | *domEx1279-1280-1281[mec-3p::cmk-1(K71A/R74Q/R77S)::mNG, unc-122p::RFP];[unc-68(dom13)]V* | Expression of CMK-1(K71A/R74Q/R77S)::mNG in FLP in a *unc-68* gain-of-function background |
| Genetic reagent (C. elegans) | DAG1282-1283-1284 | This study | *domEx1282-1283-1284[mec-3p::cmk-1(W305S)::mNG, unc-122p::RFP];[unc-68(dom13)]V* | Expression of CMK-1(W305S)::mNG in FLP in a *unc-68* gain-of-function background |
| Genetic reagent (C. elegans) | DAG1436-1437-1438 | This study | *domEx1436-1437-1438[cmk-1p::cmk-1(wt)::mNG, unc-122p::GFP]; domIs272[mec-3p::QF, mec-4p::QS, QUAS::CoCHR, unc-122p::RFP] II; cmk-1(syb1375) IV* | Rescue of *cmk-1(syb1375)* using *cmk-1* promoter, in the *[FLP::CoChR]* optogenetic background |
| Genetic reagent (C. elegans) | DAG1439-1440-1441 | This study | *domEx1439-1440-1441[mec-3p::cmk-1(wt)::mNG, unc-122p::GFP]; domIs272[mec-3p::QF, mec-4p::QS, QUAS::CoCHR, unc-122p::RFP] II; cmk-1(syb1375) IV* | Rescue of *cmk-1(syb1375)* using *mec-3* promoter, in the *[FLP::CoChR]* optogenetic background |
| Genetic reagent (C. elegans) | DAG1322-1323-1324 | This study | *domEx1322-1323-1324[ttx-1p::QF, QUAS::cmk-1(wt)::mNG, unc-122p::RFP]* | Expression of CMK-1(wt)::mNG in AFD |
| Genetic reagent (C. elegans) | DAG1325-1326-1327 | This study | *domEx1325-1326-1327[ttx-1p::QF, QUAS::cmk-1(K71A/R71Q/R77S)::mNG, unc-122p::RFP]* | Expression of CMK-1(K71A/R71Q/R77S)::mNG in AFD |
| Genetic reagent (C. elegans) | DAG1328-1329-1330 | This study | *domEx1328-1329-1330[ttx-1p::QF, QUAS::cmk-1(R77S)::mNG, unc-122p::RFP]* | Expression of CMK-1(R77S)::mNG in AFD |
| Genetic reagent (C. elegans) | DAG1331-1332-1333 | This study | *domEx1331-1332-1333[ttx-1p::QF, QUAS::cmk-1(V292A/V294A)::mNG, unc-122p::RFP]* | Expression of CMK-1(V292A/V294A)::mNG in AFD |
| Genetic reagent (C. elegans) | DAG1433-1434-1435 | This study | *domEx1433-1434-1435[ttx-1p::QF, QUAS::cmk-1(W305S)::mNG, unc-122p::RFP]* | Expression of CMK-1(W305S)::mNG in AFD |
| Genetic reagent (C. elegans) | DAG1469-1470-1471 | This study | *domEx1469-1470-1471[ttx-1p::QF, QUAS::cmk-1(wt)::mNG, unc-122p::RFP]; unc-68(dom13) V* | Expression of CMK-1(wt)::mNG in AFD in the *unc-68* gain-of-function background |
| Genetic reagent (C. elegans) | DAG1472-1473-1474 | This study | *domEx1472-1473-1474[ttx-1p::QF, QUAS::cmk-1(R77S)::mNG, unc-122p::RFP]; unc-68(dom13) V* | Expression of CMK-1(R77S)::mNG in AFD in the *unc-68* gain-of-function background |
| Genetic reagent (C. elegans) | DAG1475-1476-1477 | This study | *domEx1475-1476-1477[ttx-1p::QF, QUAS::cmk-1(W305S)::mNG, unc-122p::RFP]; unc-68(dom13) V* | Expression of CMK-1(W305S)::mNG in AFD in the *unc-68* gain-of-function background |
| Genetic reagent (C. elegans) | DAG1513 | This study | *domEx1513[ttx-1prom::QF, QUAS::YC2.3]* | Cameleon (YC2.3) expression in AFD for calcium imaging |
| Genetic reagent (C. elegans) | DAG616 | This study | *domEx616[mec-3p::calbindin, unc-122p::GFP]; domSi437[mec-3p::cmk-1::mNG::3xFlag::unc-543'UTR] II* | Calcium buffering in FLP via the expression of Calbindin |
| Genetic reagent (C. elegans) | AQ2145 | Gift from Bill Schafer | *ljEx19[egl-46p::YC2.3; lin15(+)]* | Cameleon (YC2.3) expression. |
| Genetic reagent (C. elegans) | DAG747 | *Saro et al., 2020* | *unc-68(r1161) V; ljEx19[egl-46p::YC2.3; lin15(+)]* | Cameleon (YC2.3) expression in FLP for calcium imaging in an *unc-68* loos-of-function *mutant background.* |

*Continued on next page*

*Continued*

| Reagent type (species) or resource | Designation | Source or reference | Identifiers | Additional information |
|---|---|---|---|---|
| Genetic reagent (*C. elegans*) | DAG792 | *Saro et al., 2020* | *egl-19(n582) IV; ljEx19[egl-46p::YC2.3; lin15(+)]* | Cameleon (YC2.3) expression in FLP for calcium imaging in *egl-19(n582) IV* mutant background |
| Genetic reagent (*C. elegans*) | DAG918 | *Saro et al., 2020* | *ocr-2(ak47) IV; ljEx19[egl-46p::YC2.3; lin15(+)]* | Cameleon (YC2.3) expression in FLP for calcium imaging in *ocr-2(ak47) IV* mutant background |
| Genetic reagent (*C. elegans*) | DAG1001 | *Saro et al., 2020* | *unc-2(ra612) X; ljEx19[egl-46p::YC2.3; lin15(+)]* | Cameleon (YC2.3) expression in FLP for calcium imaging in *unc-2(ra612) X* mutant background |
| Genetic reagent (*C. elegans*) | DAG857-859 | This study | *domSi439[mec-3p::cmk-1::mNeonGreen::3xFlag::unc-54UTR] II; domEx857-859[mec-3p::egl-13NLS::CeBFP::unc-54UTR]* | Expression of a blue nuclear marker in FLP already expressing CMK-1::mNG green |
| Genetic reagent (*C. elegans*) | DAG933-934 | This study | *domEx933-934[mec-3p::QF]; [QUAS::mNeonGreen, unc-122p::RFP]; [mec-3p::egl-13NLS_CeBFP::unc-54UTR]* | Expression of a blue nuclear marker in FLP already labeled in green |
| Genetic reagent (*C. elegans*) | DAG1650 | This study | *domEx1650[mec-3p::Calb28K::unc54UTR]; [mec-3p::QF::UTR54]; [QUASp::YC2.3::UTR54]* | Cameleon (YC2.3) expression in FLP for calcium imaging in presence of Calbindin |
| Genetic reagent (*C. elegans*) | DAG1652-1653-1654 | This study | *domEx1652-1653-1654[mec-3p::rParv::unc-54UTR]; domSi439[mec-3p::cmk-1::mNeonGreen::3xFlag::unc-54UTR] II* | Calcium buffering in FLP via the expression of Parvalbumin |
| Genetic reagent (*C. elegans*) | DAG1655-1656-1657 | This study | *domEx1655-1656-1657[mec-3p::rParv_mutant::unc-54UTR]; domSi439[mec-3p::cmk-1::mNeonGreen::3xFlag::unc-54UTR] II* | Negative control for calcium buffering in FLP via the expression of mutant Parvalbumin (K92V, D93A, D95A, K97V, and E100V) |

## *C. elegans* strains and growth conditions

*C. elegans* strains used in this study are reported in the Key resources table. All strains were grown as previously described *Stiernagle, 2006* on nematode growth media (NGM) plates with OP50 *E. coli*, at 20°C (unless otherwise stated). For optogenetic experiments, we used NGM plates containing all-*trans*-retinal (ATR), as well as regular NGM plates as control. ATR plates were prepared by adding 0.1% (v/v) of ATR stock (100 mM, in ethanol) to the OP50 bacteria suspension prior to seeding. 300 µl of this mix was used to seed 6 cm plates, containing 8 ml of NGM.

## Transgenesis

Plasmid DNA was purified with the GenElute HP Plasmid miniprep kit (Sigma) and microinjected in the worm gonad according to a standard protocol (*Evans, 2006*). We used *unc-122p::GFP/RFP* as co-injection markers to identify transgenic animals.

## Promoter plasmids (Multi-Site Gateway Slot 1)

Entry plasmids containing specific promoters were constructed by PCR from N2 genomic DNA, with primers flanked with attB4 and attB1r recombination sites and cloned into pDONR-P4-P1R vector (Invitrogen) by BP recombination. Plasmids and primer sequences are reported in the supplementary information, *Appendix 1—table 1*.

## Coding sequence plasmids (Multi-Site Gateway Slot 2)

Entry plasmids containing specific coding DNA sequences were constructed by PCR from N2 cDNA with primers flanked with attB1 and attB2 recombination sites and cloned into pDONR_221 vector (Invitrogen) by BP recombination. Plasmids and primer sequences are listed in the supplementary information, *Appendix 1—table 1*.

## Site-directed mutagenesis

All the deletions and point mutations were generated by inverse PCR-based site-directed mutagenesis (*Hemsley et al., 1989*). In brief, whole plasmids (entry plasmids containing *cmk-1* coding DNA

sequences) were amplified with the KOD Hot Start DNA Polymerase (Novagen; Merck). Primers were phosphorylated in 5′ and were designed to contain the desired point mutation(s) and to hybridize in a divergent and back-to-back manner on the plasmid. After electrophoresis, linear PCR products were purified from agarose gel (1%) with a Zymoclean-Gel DNA Recovery kit (Zymo Research) and circularized with DNA Ligation Kit <Mighty Mix> (Takara). The plasmid templates, primer sequences, and names of resulting mutation-carrying plasmids are reported in *Appendix 1—table 1*.

## Imaging of fluorescent CMK-1 reporter protein

### Worm preparation for CMK-1 imaging in FLP

Worms were synchronized according to standard procedure with hypochlorite treatment and grown at 20°C. First-day adult animals were collected from NGM bacterial plates with distilled water, transferred to 1.5 ml microcentrifuge tubes and washed once with distilled water. 20 µl of a dense worm suspension were transferred to PCR tubes and incubated in a thermocycler at 20 or 28°C for 1.5 hr unless otherwise stated. Prior to imaging, worms were immobilized with the addition of $NaN_3$ (final concentration 1% m/v), transferred on a glass slide and covered with a coverslip. Imaging was carried out during the next 5 min. For leptomycin B treatment, it was used at a final concentration of 100 ng/µl. It was added from a stock at 5 µg/µl diluted in M9 buffer with 10% (v/v) ethanol. Control experiment included only the vehicle.

### Worm preparation for CMK-1 imaging in AFD

Worms were prepared as described above for FLP imaging, except that they were grown on NGM plates at either 15 or at 25°C prior to collection, and scored immediately (without incubation in a thermocycler).

### Microscopy

FLP and AFD images to measure the nuclear/cytoplasmic ratio were acquired in a Zeiss Axioplan2 fluorescence microscope, with a 40× (air, NA = 0.95) objective and constant illumination parameters.

### Replicates

At least two independent transgenic lines (in most cases three lines, see Key resources table) were scored for each genotype, each on at least three different experimental days. Wild-type control was systematically run in parallel.

## Determination of CMK-1 nuclear/cytoplamic ratio

For CMK-1 subcellular localization analysis, the intensity of fluorescence was first measured for each neuron in three regions of interest (ROIs): Nucleus, Cytoplasm, Background. The nuclear/cytoplasmic ratio was calculated as (Nucleus − Background)/(Cytoplasm − Background). A ratio >1 indicates a nuclear accumulation of CMK-1, while a value <1 a cytoplasmic biased ratio. All three ROIs were ellipses of the same area. The Background ROI was defined in a worm region close to the neuron to take autofluorescence into account. Nuclear and Cytoplasmic ROIs were defined based on the mNG green signal and the shape of the neuron, via a procedure that was first validated with the use of a second fluorescent nuclear marker (nuclear ceBFP::NLS). We noticed that the nucleus was always laying very centrally, occupying a large part of the cell body (*Figure 1—figure supplement 3*). There is actually no possibility for the nucleus to fit in any narrower region of the cell. So even in situations where the green fluorescence signal is diffuse and does not itself reveal the localization of the nucleus, it is possible to make reliable predictions. The nuclear ROI was defined in the middle of the cell body (where it is the widest) and the cytoplasmic ROI decentered to a narrower region of the cell (*Figure 1—figure supplement 3*). We validated this approach by a blind test, in which we acquired a series of images with diffuse mNG (homogenously distributed between the nucleus and the cytoplasm) and nuclear BFP, and defined the ROI solely based on the green channel. After an a posteriori verification with the blue channel, we found that >98% of the ROI pairs (69/70) were correctly defined. Of note, when working with CMK-1::mNG, a nuclear accumulation or depletion of signal is often seen and further ascertains the localization of the nucleus. Therefore, the very rare errors caused by the mis-definition of ROIs will occur only when the nuclear and the cytoplasmic signals are similar; a situation where the ratio value will anyway be very close to 1, regardless of where the ROIs are defined.

Based on this assessment, we concluded that our method is valid to score CMK-1:.mNG subcellular localization, without the need for a systematic use of a nuclear co-marker.

### gcy-8p reporter imaging and quantification

AFD images to measure *gcy-8p::GFP* transcriptional reporter expression were acquired and processed like described above for FLP, except that only two ROIs were defined (Cell and Background). The expression value was calculated as Cell – Background.

### ima-3p reporter imaging

For *ima-3p* reporter imaging, a Leica TCS SPE-II confocal microscope equipped with 488 nm and 532 nm wavelength diode lasers was used with an ACS APO 40× (oil, NA = 1.15) objective.

## Expression and purification of IMA-3-HIS6, CMK-1(wt), and CMK-1(K71A/R74Q/R77S)-GST

DNA encoding proteins of interest were PCR amplified and cloned into NdeI and BamHI restriction sites of pDK2409 for the GST-TEV-tagged protein and of pDK2832 for the His6-tagged proteins. Plasmids were transformed into *E. coli* BL21 (DE3) (NEB). The proteins were purified after 5 hr of induction with 0.5 mM IPTG at 24–25°C. Cells were collected and lysed in a Microfluidizer Processor M-110L. The cell lysate clean supernatant was incubated for 2 hr with nickel-nitrilotriacetic acid beads (Ni-NTA-Qiagen, Hilden, Germany) as per the manufacturer's instructions for IMA-3-His6 and with Glutathione superflow beads (Qiagen) for GST-tagged CMK-1 variants.

Ni-NTA beads binding IMA-3-His6 were washed three times in imidazole gradient and eluted in 1 mM PMSF, 0.1% NP-40, 500 mM imidazole, pH 8.0. The elution of CMK-1 was done by incubating the beads with 2% TEV enzyme in lysis buffer (50 mM Tris-HCl pH 7.5, 150 mM NaCl, 1.5 mM MgCl$_2$, 5% glycerol) with 1 mM PMSF, 0.1% NP-40, 1 mM DTT. This step cleaves the GST tag to recover untagged CMK-1. For MST experiments, IMA-3-His6 was applied to Zeba desalting spin columns (Thermo Scientific) to remove the excess of imidazole and the buffer was exchanged to the same buffer of CMK-1-GST with 1 mM PMSF and 0.1% NP-40. Protein concentration was determined by Pierce Microplate BCA protein assay Kit-Reducing Agent Compatible (Thermo Scientific) using BSA as protein standard.

## Binding affinity quantification by MST

MST experiments were performed using a Monolith NT.115 from NanoTemper Technologies to assess the affinity between IMA-3 and CMK-1(wt) or CMK-1(K71A/R74Q/R77S). IMA-3-HIS6 was labeled using the RED-tris-NTA His tag protein labeling kit, resuspended in HEPES 15 mM, 0.05% Tween 20. Labeled IMA-3-HIS6 protein (at a concentration of 100 nM) was added to a serial dilution of unlabeled CMK-1 prepared in binding buffer (1× PBS with 0.05% Tween 20). Samples were loaded into MST standard capillaries Monolith NT.115 MO-k022 (NanoTemper) and MST measurements were performed using 40% laser power setting. For the assays in the presence of Ca$^{2+}$/CaM, we added 2 mM CaCl$_2$ and 8 µM CaM (from bovine brain; high purity; Sigma).

The dissociation constants *Kd* were obtained by plotting the normalized fluorescence (Fnorm) against the logarithm of ligand concentration and fitting using the *Kd* model with the MO-Affinity Analysis software (NanoTemper Technologies). Experiments were performed in triplicates. Statistical comparisons between *Kd* values were made using the method described by *Paternoster et al., 1998*.

## Behavioral assays

All experimental replicates were obtained over at least three different days. Adult worms were either synchronized by standard hypochlorite treatment or picked as L4 larvae on NGM plates 1 day before the experiments.

### Heat-evoked reversal

Worm populations (n ≥ 50 animals) crawling on food-free NGM-plates were exposed to two 4 s heat pulses with a 20 s recovery period in between. The first pulse was at a heating power of 0.3 W/m$^2$ and the second one at a heating power of 0.6 W/m$^2$. We used a previously described system for heat

delivery, worm movie recording, and reversal flagging (*Lia and Glauser, 2020*). The genotype effects were the same at both heating power levels, and the data were pooled for the reversal analysis.

## Thermotaxis

Well-fed animals were tested in thermotaxis assays as previously described (*Ramot et al., 2008a*). Briefly, worms were grown at 20°C, recovered from the plate in distilled water, washed twice and spread at the center of a rectangular plate pre-equilibrated to form a linear thermogradient gradient (~1°C/cm). The starting temperature was 23°C. We ensured homogenous developmental and feeding states across cultivation plates by adjusting animal density such that food was constantly available during their development.

## Noxious heat thermogradient assays

The noxious heat thermogradient assays, in which we scored the worm dispersal in a temperature gradient from 29 to 37°C, were performed as previously reported. Like in previous studies, assays were performed with animals who had been starved for 5–7 hr in order to eliminate the contribution of the thermotaxis circuit. For the heat-evoked desensitization experiments, we incubated the plates at 28°C during the last 1.5 hr of starvation. Instead of using a heat-avoidance index as in previous studies, we calculated the temperature of the third quartile of the worm distribution because we found that it was a more robust indicator of the worm spreading on the thermal gradient. Indeed, over the conditions considered for the present study, the relative error of the heat-avoidance index was 22% on average (range 15–32%) versus 8% on average (range 4–14%) for the third quartile-based metrics.

## Optogenetics

Optogenetic analysis was carried out in a previously described [*FLP::CoChR*] genetic background (*Schild and Glauser, 2015*), in which blue light stimuli of low intensities can activate FLP and trigger reversal responses.

## Light-evoked reversal assay

Single forward-moving animals were illuminated with blue light during 0.5 s. Scoring was done manually and any backward movement taking place during the stimulation was counted as a positive response, as previously described (*Marques et al., 2019*). Animals were stimulated three times in a row, leaving enough time for forward locomotion to restart in between stimuli. We did not observe a significant habituation effect over the three trials and the whole pool of trials was used for statistical analyses. Controls run in the absence of ATR, as well as with red light stimuli, showed no light dose–response effects (*Figure 5—figure supplement 1*).

## Heat adaptation prior to light-evoked reversal assays

Young adults were washed from '+ ATR' NGM plates like described for CMK-1 imaging, and incubated for 90 min in a thermocycler at 20°C (naïve) and 28°C (adapted). After incubation, worms were transferred on bacteria-free NGM dishes, pre-equilibrated at 20 or 28°C, and left to crawl for 10 min before the experiment.

## Calcium imaging

## Calcium imaging in AFD

To assess intracellular calcium in AFD, we maintained [*ttx-1p::YC2.3*] worms at 15 or 25°C, overnight. Adult worms were prepared as described in *Saro et al., 2020* and imaged using an inverted epifluorescence microscope (Leica DMI6000B) equipped with a HCX PL Fluotar L40x/0.60 CORR dry objective, a Leica DFC360FX CCD camera (1.4 M pixels, 20 fps), an EL6000 Light Source, and a fast filter wheel for FRET imaging. The recording and calcium imaging analysis were performed as in *Saro et al., 2020*. Data are reported as YFP/CFP ratios with no baseline normalization in order to enable the comparison between resting calcium levels across animals and conditions. Animals were selected based on the overall fluorescence level of the reporter in order to have similar expression levels across conditions, which we verified a posteriori. This verification was made by summing the CFP and YFP emission signals (after excitation at 405 nm) in order to obtain a metrics representing

the total fluorescence independently of varying FRET levels. This metrics was not significantly different between the sets of traces recorded at 15 and 25°C, respectively (15°C: average = 517, sem = 93; 25°C, average = 552, sem = 80; arbitrary units; p=0.73 by Student's t-test).

## Calcium imaging in FLP
To assess intracellular calcium in the calbindin-expressing transgenic animals and in the mutants affecting different calcium channels, we incubated synchronized [*egl-46p::YC2.3*] adult worms at 20 or 28°C for 1.5 hr.

### Statistical tests
Comparisons were made with one-way and two-way ANOVAs using Jamovi (The jamovi project (2021), jamovi (Version 1.6) [Computer Software]; retrieved from https://www.jamovi.org). A visual inspection of Q-Q plots and nonsignificant Kolmogorov–Smirnov tests ($p > 0.01$) suggested that all datasets could be considered to follow a normal distribution. However, some datasets returned significant results with the Shapiro–Wilk test ($p < 0.01$). For that reason, we conducted and reported the results of both regular ANOVAs and robust ANOVAs. Of note, the two methods gave similar results. Comparisons giving significant effects ($p < 0.01$) with ANOVAs were followed by Bonferroni post-hoc tests.

## Acknowledgements

We are grateful to Lisa Schild and Laurence Bulliard for expert technical support, to Filipe Marques and Andrei-Stefan Lia for advices on behavioral analyses, to Martina Rudgalvyte as well as to Dieter Kressler from the Metabolomics and Proteomics Platform (MAPP, University of Fribourg) for help with the recombinant protein expression and purification, to Lola Hostettler and Laurie Zbinden for help with the construction of some plasmids, to Marc Hammarlund, Bill Schafer, Piali Sengupta, Miriam Goodman, and Chiou-Fen Chuang for the gift of plasmids and strains, and to Boris Egger from the Bioimage facility (University of Fribourg) for assistance with microscopy. We furthermore thank Miriam Goodman for insightful comments on an earlier version of the article. Some strains were provided by the CGC (RRID:SCR_007341), which is funded by NIH Office of Research Infrastructure Programs (P40 OD010440). The study was supported by the Swiss National Science Foundation (BSSGI0_155764, PP00P3_150681, and 310030_197607 to DAG).

## Additional information

### Funding

| Funder | Grant reference number | Author |
| --- | --- | --- |
| Schweizerischer Nationalfonds zur Förderung der Wissenschaftlichen Forschung | BSSGI0_155764 | Dominique A Glauser |
| Schweizerischer Nationalfonds zur Förderung der Wissenschaftlichen Forschung | PP00P3_150681 | Dominique A Glauser |
| Schweizerischer Nationalfonds zur Förderung der Wissenschaftlichen Forschung | 310030_197607 | Dominique A Glauser |

The funders had no role in study design, data collection and interpretation, or the decision to submit the work for publication.

## Author contributions
Domenica Ippolito, Formal analysis, Investigation, Methodology, Resources, Validation, Visualization, Writing – original draft, Writing – review and editing; Saurabh Thapliyal, Formal analysis, Investigation, Writing – review and editing; Dominique A Glauser, Conceptualization, Formal analysis, Funding acquisition, Project administration, Supervision, Visualization, Writing – original draft, Writing – review and editing

## Author ORCIDs
Dominique A Glauser  http://orcid.org/0000-0002-3228-7304

## Decision letter and Author response
Decision letter https://doi.org/10.7554/eLife.71443.sa1
Author response https://doi.org/10.7554/eLife.71443.sa2

## Additional files

### Supplementary files
• Transparent reporting form

### Data availability
All data generated or analysed during this study are included in the manuscript and supporting files. Source data files have been provided for Figures 1, 2, 3, 4, 5 and 6. The article does not include any large dataset.

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

# Appendix 1

Plasmids and primers used for cloning are presented in *Appendix 1—table 1*.

**Appendix 1—table 1.** Plasmids and primers used for cloning.

| Plasmid types | Plasmid name, cloning, and primer information |
|---|---|
| Promoter plasmids (Multi-Site Gateway slot 1): | **dg701***[slot1 Entry ima-3p]*<br>attB4ima-3_F: ggggacaagtttgtacaaaaaagcaggcttacatatgagttcaaacagacaggcttatt<br>attB1ima-3_R: ggggaccactttgtacaagaaagctgggtcggatcccttttccaaagttccatcctccg<br>**dg508***[slot1 Entry ttx-1p]*<br>attB4_F: ggggacaactttgtatagaaaagttgatccatactcaggggaacagtgt<br>attB1_R: ggggactgctttttgtacaaacttgtgaagcaggaatatatgacaaatgaaatacg<br>The generation of **dg68***[slot1 Entry mec-3p(noATG)]* and **dg229***[slot1 Entry QUASprom]* was previously described in ***Schild and Glauser, 2015*** and the generation of **mg268***[slot1 Entry cmk-1p]* in ***Schild et al., 2014***. |
| Coding sequence plasmids (Multi-Site Gateway slot 2): | **dg703***[slot2 Entry ima-3cds]*<br>attB1ima3cds_F: ggggacaagtttgtacaaaaaagcaggcttacatatgagttcaaacagacaggcttatt<br>attB2ima3cds_R: ggggaccactttgtacaagaaagctgggtcggatcccttttccaaagttccatcctccg<br>The generation of **dg240** *[slot2 Entry QF_withATG]* and **dg245***[mec-4p(2kb)::QS::SL2mCherry]* was previously described in ***Schild and Glauser, 2015***; the generation of **dg286** *[slot2 Entry (cmk-1cds_noSTOP)]* was described in ***Hostettler et al., 2017***; the generation of **dg651** *[slot2 Entry egl-13NLS::wrmScarlet]* was described in ***Marques et al., 2019***; the generation of **dg718***[slot2 Entry YC2.3]* was described in ***Saro et al., 2020***.<br>**dg215***[pENTR slot2_calb28K]* created by PCR amplification from [*odr-3p::calbindin D28K*] (gift from Chiou-Fen Chuang) followed by subcloning into pDON221 via BP recombination.<br>**dg217***[pENTR slot2_rParv]* created by PCR amplification from [*odr-3p:: rParv*] (gift from Chiou-Fen Chuang) followed by subcloning into pDON221 via BP recombination.<br>**dg219***[pENTR slot2_rParv_mutant]* created by PCR amplification from [*odr-3p:: rParvCDEF/AV*] (gift from Chiou-Fen Chuang) followed by subcloning into pDON221 via BP recombination.<br>**dg643***[pENTR slot2 Ex13_CeBFP-reporter_V2]* was created amplifying by PCR and subcloning into pDON221 by BP recombination a codon-optimized CeBFP version containing three artificial introns obtained via gene synthesis (Eurofins DNA).<br>**dg650***[slot2 Entry egl-13NLS::wrmScarlet]* was created adding to dg643 the egl-13NLS by PCR with the following primers:<br>NLSRightHalf_CeBFP_F: aaacgcgaagaagcttgccaaggaagttgaaaatggatccatgtcagagcttattaagg<br>NLSLeftHalf_CeBFP_R: cactcagttttgtcggattcgcttttcgtctacggctcatgttgctagcggtacctaag |
| Plasmids carrying *cmk-1* coding sequence with small deletions and primers used to introduce these deletions: | **dg168***[slot2 Entry cmk-1(Δ315–323)]*<br>cmk-1_324F: tcctcaaatagcaatcgcctacagaaac<br>cmk-1_314R: tgctgcggggctgcgtt<br>**dg173***[slot2 Entry cmk-1(Δ318–324)]*<br>cmk-1_325F: tcaaatagcaatcgcctacagaaacaag<br>cmk-1_317R: ctggcggattgctgcgg<br>**dg192***[slot2 Entry cmk-1(Δ288–294)]*<br>CMK-1_287R: atcgtgtgtgtacgccgtatttc<br>CMK-1_295F: catcttaagaagagtttggcaaaacgga |

*Appendix 1—table 1 Continued on next page*

*Appendix 1—table 1 Continued*

| Plasmid types | Plasmid name, cloning, and primer information |
|---|---|
| | **dg588**[*slot2 Entry cmk-1(K71A/R74Q/R77S)*]<br>NLS$^{71-78}$_F: cagaagctctcacacaacaatattgttcaactattcga<br>NLS$^{71-78}$_R: cagaactgcaatctcgttttccagtgattcttc<br>**dg589**[*slot2 Entry cmk-1(W305S)*]<br>W305S_F: ccaaaaaggcttacaacgcagcc<br>W305S_R: agttccgttttgccaaactcttct<br>**dg590**[*slot2 Entry cmk-1(V292A/V294A)*]<br>NES$^{288-294}$_F: cgctcatcttaagaagagtttggcaaaacgga<br>NES$^{288-294}$_R: gcggcagttccgtgaatatcgtgtgtgtac<br>**dg591**[*slot2 Entry cmk-1(L321A/L323A)*]<br>NES$^{314-323}$_F: gctcgtgcttcctcaaatagcaatcgcctacagaa<br>NES$^{314-323}$_R: catttgaagctggcggattgct<br>**dg593**[*slot2 Entry cmk-1(K307Q)*]<br>NLS$^{297-308}$_F: caggcttacaacgcagccgc<br>NLS$^{297-308}$_R: tttccagttccgtttttgccaaactcttc<br>**dg612**[*slot2 Entry cmk-1(R303S/K307Q)*]<br>NLS$^{297-308}$_F: tccaactggaaacaggcttacaacg<br>NLS$^{297-308}$_R: ttttgccaaactcttcttaagatgtacg<br>**dg658**[*slot2 Entry cmk-1(K71A/R74Q/R77S/V292A/V294A)*]<br>NLS$^{71-78}$_F/R with dg608 as template<br>**dg663**[*slot2 Entry cmk-1(V292A/V294A/W305S)*]<br>W305S_F/R with dg608 as template<br>**dg674**[*slot2 Entry cmk-1(K71A)*]<br>NLS$^{71-78}$_ K71A_F: aggaagctccgacacaac<br>NLS$^{71-78}$_ K71A_R: cagaactgcaatctcgttttccagtgattcttc<br>**dg675**[*slot2 Entry cmk-1(R74Q)*]<br>NLS$^{71-78}$_ R74Q_F: cagaagctccgacacaacaatattgttcaacta<br>NLS$^{71-78}$_ R74Q_R: cagaactttaatctcgttttccagtgattcttc<br>**dg676**[*slot2 Entry cmk-1(R77S)*]<br>NLS$^{71-78}$_ R77S_F: tcacacaacaatattgttcaactattcga<br>NLS$^{71-78}$_ R77S_R: gagcttcctcagaactttaatctc<br>**dg688**[*slot2 Entry cmk-1(I315A)*] |
| Plasmids with *cmk-1* point mutations and primers used for site-directed mutagenesis: | NES$^{314-323}$_315F: ctccgtctctcctcaaatagcaatcgc<br>NES$^{314-323}$_315R: catttgaagctggcgggctgctgcg<br>**dg689**[*slot2 Entry cmk-1(I315A/L321A/L323A)*]<br>NES$^{314-323}$_315F/R with dg608 as template |
| 3' UTR and tagging plasmids (Multi-site Gateway slot 3): | **mg277**[*slot3 Entry SL2::mCherry*] was previously described in ***Schild et al., 2014***.<br>**mg211**[*slot3 Entry unc-54 3'UTR*] (aka pMH473) was a gift from Marc Hammarlund.<br>**dg397**[*slot3 Entry mNG::3xFLAG::unc-54 3'UTR*] previously described in ***Hostettler et al., 2017***. |
| Selection markers used for transgenesis: | **dg9** [coel::RFP] (*or unc-122p::RFP*) was a gift from Piali Sengupta (Addgene plasmid # 8938).<br>**dg396** [coel::GFP] (*or unc-122p::GFP*) was a gift from Piali Sengupta (Addgene plasmid # 8937). |

*Appendix 1—table 1 Continued on next page*

Appendix 1—table 1 Continued

| Plasmid types | Plasmid name, cloning, and primer information |
|---|---|
| | **dg405**[mec-3p::cmk-1::mNG::3xFlag] was previously created through an LR recombination reaction between dg68, dg286, dg397 and PCFJ150 (**Hostettler et al., 2017**). |
| | **dg425**[cmk-1p::cmk-1(wt)::mNG::3xFlag::unc-54 3'UTR] was previously created through an LR recombination reaction between mg268, dg286, dg397, and pDEST-R4-P3 (**Hostettler et al., 2017**). |
| | **dg515**[mec-3p::cmk-1(Δ315–323)::mNG::3xFlag::unc-54 3'UTR] was created through an LR recombination reaction between dg68, dg168, dg397, and pDEST-R4-P3. |
| | **dg520**[mec-3p::cmk-1(Δ318–324)::mNG::3xFlag::unc-54 3'UTR] was created through an LR recombination reaction between dg68, dg173, dg397, and pDEST-R4-P3. |
| | **dg524**[mec-3p::cmk-1(Δ288–294)::mNG::3xFlag::unc-54 3'UTR] was created through an LR recombination reaction between dg68, dg192, dg397, and pDEST-R4-P3. |
| | **dg605**[mec-3p::cmk-1(K71A/R74Q/R77S)::mNG::3xFlag::unc-54 3'UTR] was created through an LR recombination reaction between dg68, dg588, dg397, and pDEST-R4-P3. |
| | **dg606**[mec-3p::cmk-1(W305S)::mNG::3xFlag::unc-54 3'UTR] was created through an LR recombination reaction between dg68, dg589, dg397, and pDEST-R4-P3. |
| | **dg607**[mec-3p::cmk-1(V292A/V294A)::mNG::3xFlag::unc-54 3'UTR] was created through an LR recombination reaction between dg68, dg590, dg397, and pDEST-R4-P3. |
| | **dg608**[mec-3p::cmk-1(L321A/L323A)::mNG::3xFlag::unc-54 3'UTR] was created through an LR recombination reaction between dg68, dg591, dg397, and pDEST-R4-P3. |
| | **dg610**[mec-3p::cmk-1(K307Q)::mNG::3xFlag::unc-54 3'UTR] was created through an LR recombination reaction between dg68, dg593, dg397, and pDEST-R4-P3. |
| | **dg613**[mec-3p::cmk-1(R303S/K307Q)::mNG::3xFlag::unc-54 3'UTR] was created through an LR recombination reaction between dg68, dg612, dg397, and pDEST-R4-P3. |
| | **dg665**[mec-3p::cmk-1(K71A/R74Q/R77S/V292A/V294A)::mNG::3xFlag::unc-54 3'UTR] was created through an LR recombination reaction between dg68, dg658, dg397, and pDEST-R4-P3. |
| | **dg670**[mec-3p::cmk-1(V292A/V294A/W305S)::mNG::3xFlag::unc-54 3'UTR] was created through an LR recombination reaction between dg68, dg663, dg397, and pDEST-R4-P3. |
| | **dg677**[mec-3p::cmk-1(K71A)::mNG::3xFlag::unc-54 3'UTR] was created through an LR recombination reaction between dg68, dg674, dg397, and pDEST-R4-P3. |
| | **dg678**[mec-3p::cmk-1(R74Q)::mNG::3xFlag::unc-54 3'UTR] was created through an LR recombination reaction between dg68, dg675, dg397, and pDEST-R4-P3. |
| | **dg679**[mec-3p::cmk-1(R77S)::mNG::3xFlag::unc-54 3'UTR] was created through an LR recombination reaction between dg68, dg676, dg397, and pDEST-R4-P3. |
| | **dg694**[mec-3p::cmk-1(I315A)::mNG::3xFlag::unc-54 3'UTR] was created through an LR recombination reaction between dg68, dg688, dg397, and pDEST-R4-P3. |
| | **dg695**[mec-3p::cmk-1(I315A/L321A/L323A)::mNG::3xFlag::unc-54 3'UTR] was created through an LR recombination reaction between dg68, dg689, dg397, and pDEST-R4-P3. |
| | **dg705**[ima-3p::egl-13NLS::wrmScarlet::unc-54 3'UTR] was created through an LR recombination reaction between dg701, dg651, mg211, and pDEST-R4-P3. |
| | **dg876**[QUAS::cmk-1(wt)::mNG::3xFlag::unc-54 3'UTR] was created through an LR recombination reaction between dg229, dg286, dg397, and pDEST-R4-P3. |
| | **dg877**[QUAS::cmk-1(K71A/R74Q/R77S)::mNG::3xFlag::unc-54 3'UTR] was created through an LR recombination reaction between dg229, dg588, dg397, and pDEST-R4-P3. |
| | **dg878**[QUAS::cmk-1(R77S)::mNG::3xFlag::unc-54 3'UTR] was created through an LR recombination reaction between dg229, dg286, dg676, and pDEST-R4-P3. |
| | **dg879**[QUAS::cmk-1(V292A/V294)::mNG::3xFlag::unc-54 3'UTR] was created through an LR recombination reaction between dg229, dg590, dg397, and pDEST-R4-P3. |
| | **dg882**[QUAS::cmk-1(W305S)::mNG::3xFlag::unc-54 3'UTR] was created through an LR recombination reaction between dg229, dg589, dg397, and pDEST-R4-P3. |
| | **dg883**[ttx-1p::QF::unc-54 3'UTR] was created through an LR recombination reaction between dg508, dg240, mg211, and pDEST-R4-P3. |
| | **dg777**[QUAS::YC2.3::unc-54 3'UTR] was created through an LR recombination reaction between dg229, dg718, mg211 , and pDEST-R4-P3. |
| | **dg280**[mec-3p::Calb28K::unc54UTR] was created through an LR recombination reaction between dg68, dg215, mg211, and pDEST-R4-P3 |
| | **dg282**[mec-3p::rParv::unc-54UTR] was created through an LR recombination reaction between dg68, dg217, mg211, and pDEST-R4-P3. |
| | **dg284**[mec-3p::rParv_mutant::unc-54UTR] was created through an LR recombination reaction between dg68, dg219, mg211, and pDEST-R4-P3. |
| Expression plasmids used for transgenesis with a description of their creation: | **dg373**[QUAS::NeonGreen] was created through an LR recombination reaction between dg229 (pENTR_slot1_QUASprom) dg353, mg211, and pDEST-R4-P3. |
| | **dg653**[mec-3p::egl-13NLS::CeBFP::unc-54 3'UTR] was created through an LR recombination reaction between dg68, dg650, mg211, and pDEST-R4-P3. |
| Plasmids used for recombinant protein expression: | **dg918**[ima-3::His-6] was created by inserting *ima-3* coding sequence using NdeI and BamHI restriction sites in pDK2832 (Gift from Dieter Kressler). The *ima-3* coding sequence was obtained from a modified dg703 plasmid in which an internal BamHI site had been removed using the following primers: Ima-3Bamless_F: cgatccaaacttgcaatttgaagctg Ima-3Bamless_R: gtgctggacaagcattgaacg |
| | **dg922** [cmk-1(wt)::GST-TEV] was created by insertion of *cmk-1* cds using NdeI and BamHI restriction sites in pDK2409 (Gift from Dieter Kressler). |
| | **dg929**[cmk-1(K71A/R74Q/R77S)::GST-TEV] was created by PCR site-directed mutagenesis from dg922. |

