## [Editor Report]

This work elucidates the molecular mechanism of CaMKI shuttling between nucleus and cytoplasm and its function in thermal memory and thermal avoidance behavior in *C. elegans*. The authors thereby establish a direct link between the state of a signal transduction pathway, neuronal activity, and a complex behavioral output.

---

## [Decision Letter]

**Decision letter after peer review:**

Thank you for submitting your article "Ca^2+^/CaM binding to CaMKI promotes IMA-3 importin binding and nuclear translocation in sensory neurons: a key mechanism in behavioral adaptation" for consideration by *eLife*. Your article has been reviewed by 3 peer reviewers, one of whom is a member of our Board of Reviewing Editors, and the evaluation has been overseen by Richard Aldrich as the Senior Editor. The following individual involved in review of your submission has agreed to reveal their identity: Shawn Xu (Reviewer #2).

The reviewers have discussed their reviews with one another, and the Reviewing Editor has drafted this to help you prepare a revised submission. Please find below the reviewer's comments, we find essential to be addressed for publication in *eLife*.

Essential revisions:

*Reviewer #1:*

(1) Figure 1B, right panels show the major assay of this paper leading the authors to conclude that CMK-1 relocates to the nucleus at 28°C. This does not look convincing to me, and I wonder whether the transgene gets strongly upregulated at 28°C (often seen in *C. elegans*) and that the bright dotty speckle pattern could be aggregates. This concern could be easily diluted if the authors showed higher quality, higher resolution confocal scans together with a nuclear red marker showing that the CMK-1 clusters are indeed restricted to the nucleus. Alternatively, high quality DIC images could be used to outline the nucleus (though perhaps very challenging in adult worms; a fluorescent marker might be more straight forward). It could then be discussed what these sub-nuclear structures might be, could be even very interesting.

(2) In the same vein, I wonder how objectively the authors outlined cytoplasm and nucleus in their image analyses since, they seemed to have done this manually. Moreover, the reporter strain only expresses CMK-1::mNG and no other independent nuclear reference marker is used. Therefore, they seemed to have used the CMK-1::mNG itself to define cytoplasm and nucleus, a strategy that could mislead the authors if the CMK-1 transgene upregulates and/or aggregates at 28°C. Again, this can be easily addressed. I think they need to validate the assays with the additional high-res scans and a nuclear reference marker. Show in higher zoom images how cytoplasm and nucleus ROIs were defined for the standard assays as well.

(2) Figures 2 and 3 – some data appears on Figure 3 that appears also in Figure 2. For example, the nuclear localization of CMK-1(wt) with and without the unc-68¬ background at 20 and 28 degrees (Figure 2B, Figure 3D). Is it the same experiment or independent data? This should be clarified and if data are re-used in different figure this should be clearly stated in legends.

(3) Figure 2 and 3 – Some genetic backgrounds are used that are claimed to change the intracellular Ca++ levels in the FLP neuron with reference to the literature. Since the authors have the capacity to conduct Ca+ imaging in the FLP neuron, using a radiometric indicator, validating the expected effect of unc-68(dom13) (increased) and mec-3p::Calbindin (decreased) on intracellular free Ca++ levels in FLP would strengthen the claims of the authors. In the same vein, the various Ca++ channels mutants that do not affect localization should therefore not affect intracellular free Ca++ levels, which might be surprising. I think the correlation of CMK-1 localization with free Ca++ should be further validated in this way.

(4) Figure 6H-I and Page 24 lines 570-573: The authors refer to the result that cmk-1 mutants fail to express gcy-8 in the AFD neuron. Both cmk-1 NLS^71-78^ mutants and cmk-1 NES^288-294^ mutants did not increase the expression level of gcy-8 when grown at 25deg. However, these results do not fit the other results presented in the manuscript. In figure 6A-F the AFD neuron is shown to exhibit higher Ca+ levels and nuclear localization of CMK-1, when worms are cultivated at lower temperatures and that mutating the NLS^71-78^ lead to cytoplasmic localization of CMK-1. Figure 6H shows that at lower temperatures where CMK-1 is nuclear, the expression of gcy-8 is reduced. Therefore, one would expect that NES^288-294^ (cmk(syb1375)) mutants that increase nuclear localization of CMK-1 would have lower gcy-8 but the results show the opposite. The authors should discuss a detailed model of cmk-1 dependent gcy-8 expression to address this conflict.

(5) Figure 4 in vitro assay of IMA-3 and CMK-1 binding are comparing the effect adding Ca++ plus CaM, but I miss a Ca++ only control.

(6) Pages 9-10 and Figure 2. The authors show that CMK-1 nuclear localization is unaffected in mutants backgrounds of the genes unc-13 and unc-31 required for synaptic transmission and neuropeptide signaling. Indeed, these are classical genetic backgrounds used to test whether an effect is cell autonomous. The authors then claim at several occasions, including in the headline of this chapter that the intra-cellular calcium control of CMK-1 localization in FLP neuron is cell autonomous. This claim is not fully supported by the data since: 1. The double mutant unc-13;unc-31 was not tested. 2. The FLP and neuron make gap junction connections to other neurons that may affect their Ca+ levels and CMK-1 localization, however this possibility was not tested (for example by testing mutants in the innexins expressed by FLP). I don't think that new experiments are necessary here but the authors should accommodate for these possibilities in their conclusions and discussions.

(7) Several results described in the work suggest that there is a Ca-dependent but NLS^71-78^ -independent pathway that is involved in the nuclear localization of CMK-1 in FLP and AFD neurons: A. in Figure 3D – the addition of the unc-86(dom13) background seem to increase nuclear localization of NLS^71-78^ mutated CMK-1 (although the effect sizes are small.). Similarly, in Figure 2B, unc-86(dom13) could also increase the nuclear localization of CMK-1 in the Ca+ binding domain mutated CMK(WS305S). B. Figure 6B – the effect of mutating the NLS^71-78^ on CMK-1 localization in AFD is not fully penetrant. This is in contrast with the effect of CMK(WS305S). I suggest the authors to test whether these effects are significant and if so discuss this more clearly.

(8) Figure 5D – negative controls of no ATR or worms without the optogenetic construct are lacking. It is possible the response is a general avoidance to light. The control is mentioned in the methods section but should be shown. It is hard to imagine that there was "no response" as worms always exert spontaneous reversal behaviors.

(9) Figure 5G and H and Pages 23 and 24 – The authors should cite the work of Satterlee et al., Curr. Biol. (2004) also here when describing the effect of cmk-1 mutants on thermosensation responses and on gcy-8 expression pattern in the AFD neuron, both examined in this work (Table 1 and Figure 4).

(10) Figure 5C – instead of doing the regular thermotaxis experiment they use their own variant on it observing the third quartile of worm distribution" and claiming in the methods section that this is a more robust readout to previous methods. This statement should be validated quantitatively.

(11) Please also report absolute YFP and CFP fluorescent levels of the ratiometric Ca++ indicator; are expression levels affected in 15deg vs 25deg, which can be often observed ion transgenes?

*Reviewer #2:*

I am happy to support its publication in *eLife*. I only have a few comments.

(12) Can the authors explain why cmk-1(syb1375) mutants did not effectively spread on the assay plates?

(13) The authors may want to include raw data points in the bar graphs such that readers will get a chance to observe the data distribution patterns and sample sizes (n numbers). This has now become an increasingly common practice. Currently, all bar graphs only show error bars without raw data points or n numbers.

(14) For FLP CMK-1::GFP translocation data, the authors only showed sample images in one place: Figure 1B. All such data were presented as bar graphs without sample images. Images are easier for readers to comprehend. It is not necessary to show sample images for all bar graphs in the main figures, as this will take up too much space, though the rest may go to supplemental figures.

(15) typos: "addressed" (line 418) and "serves" (line 625).

*Reviewer #3:*

The manuscript is well written and follows a clear experimental logic with well-designed and supportive experiments and data presentation. The data support well the conclusions and claims. On the other hand, the current methodological description and presentation of statistical analysis needs to be improved.

(16) Detailed statistical information is missing; I assume the Authors used ANOVA for multiple comparisons (not mentioned in text or figure legends) with post-hoc t-test using Bonferroni correction for multiple testing (mentioned as Bonferroni contrast) so it would be good to include a detailed description of the statistical analysis used, and report the ANOVA result, dF, and post-hoc test results with exact p-values for the data analyzed.

(17) the title "Ca^2+^/CaM binding promotes CMK-1nuclear expression…" in not correct, nuclear localization would be more appropriate.

---

## [Author Response]

Essential revisions:Reviewer #1:(1) Figure 1B, right panels show the major assay of this paper leading the authors to conclude that CMK-1 relocates to the nucleus at 28°C. This does not look convincing to me, and I wonder whether the transgene gets strongly upregulated at 28°C (often seen in *C. elegans*) and that the bright dotty speckle pattern could be aggregates. This concern could be easily diluted if the authors showed higher quality, higher resolution confocal scans together with a nuclear red marker showing that the CMK-1 clusters are indeed restricted to the nucleus. Alternatively, high quality DIC images could be used to outline the nucleus (though perhaps very challenging in adult worms; a fluorescent marker might be more straight forward). It could then be discussed what these sub-nuclear structures might be, could be even very interesting.In the same vein, I wonder how objectively the authors outlined cytoplasm and nucleus in their image analyses since, they seemed to have done this manually. Moreover, the reporter strain only expresses CMK-1::mNG and no other independent nuclear reference marker is used. Therefore, they seemed to have used the CMK-1::mNG itself to define cytoplasm and nucleus, a strategy that could mislead the authors if the CMK-1 transgene upregulates and/or aggregates at 28°C. Again, this can be easily addressed. I think they need to validate the assays with the additional high-res scans and a nuclear reference marker. Show in higher zoom images how cytoplasm and nucleus ROIs were defined for the standard assays as well.

It is indeed very important that we unequivocally communicate about the nuclear re-localization of CMK-1. As CMK-1 nuclear entry is a well-documented phenomenon, which was previously published (Schild et al. 2014), we initially designed Figure 1 to only include quantitative data and it was a last-minute decision to include the representative picture. We apologize for the bad quality and poor resolution of the chosen illustration, in particular as it appeared in the assembled pdf. We have made numerous observations with epifluorescence microscopy and confocal microscopy, and we found that the signal is much more homogenous than what the previously presented picture suggested. We actually lack any evidence of clear structures in the cytoplasm or the nucleus. We have inserted a better picture set to illustrate the nuclear entry in the revised Figure 1B.

Thanks for bringing up the point regarding the definition of nucleus and cytoplasm. We realize that the description of our methodology was not clear enough regarding how we defined the regions of interest (ROI) in the nucleus and the cytoplasm, in order to calculate the signal ratio. To address this concern, we have extended the method section in the revised manuscript, (including a novel supplementary figure to illustrate our method). When developing our assay, we initially used a two-color labeling with the blue channel for a BFP nuclear marker and the green channel for CMK-1::GFP. A precise definition of the nucleus was possible, but the approach was extremely tedious and would not have been compatible with the volume of data acquisition involved in the project, given the natural variability CMK-1 localization across animals in vivo and the need for scoring thousands of animals*.* This is why we developed a procedure in which we focused solely on the green channel to define a Nuclear ROI and a Cytoplasmic ROI. These ROI were not covering the whole area of the cytoplasm or the nucleus, but were smaller ellipses (of the same area) defined in the cytoplasm and the nucleus, respectively. The FLP cell body has a slightly elongated morphology, and we noticed that the nucleus was always laying very centrally, occupying a large part of the cell body. There is actually no room to fit the nucleus toward the ‘corners’ of the cell. So even when the GFP signal is diffuse and does not reveal the localization of the nucleus, it is possible to make excellent guesses as to where one should define the ROI for the analysis: nuclear ROI in the middle of the cell body where it is the widest; cytoplasmic ROI decentered to a narrower region of the cell. We validated this approach by a blind test, in which we acquired a series of images with diffuse mNG (homogenously distributed between the nucleus and the cytoplasm) and nuclear BFP, and defined the ROI solely based on the green channel. After an a posteriori verification with the blue channel, we found that >98% of the ROI pairs (69/70) were correctly set. Of note, when working with CMK-1::mNG, a nuclear accumulation or depletion of signal is often seen and further ascertains the localization of the nucleus. Therefore, the very rare errors caused by the mis-definition of ROIs will occur only when the Nuclear/Cytoplasmic signal ratio is very close to 1; regardless of where the ROIs are defined. Based on this assessment, we conclude that our method is valid to score CMK-1:.mNG subcellular localization, without the need for a systematic use of a nuclear co-marker.

A third point raised by this reviewer relates to the potential impact of temperature on reporter expression/aggregation. We did not notice temperature-dependent aggregation, nor a flagrant increase in transgene expression after 90 min incubation at 28°C. To look in more details in a potential change in expression levels, we quantified the total green fluorescence signal over the whole FLP cell body in animals at 20°C versus 28°C (*n*=110 animals at each temperature). We found no significant expression changes in fluorescent level (20°C: average=33.3, sem=0.9; 28°C: average=35.3; sem=0.8; arbitrary units; *p* value by Student *T-*test = .11). We therefore interpret the change in fluorescence ratio reported throughout the manuscript as the result of a subcellular re-localization of CMK-1::mNG reporters.

(2) Figures 2 and 3 – some data appears on Figure 3 that appears also in Figure 2. For example, the nuclear localization of CMK-1(wt) with and without the unc-68¬ background at 20 and 28 degrees (Figure 2B, Figure 3D). Is it the same experiment or independent data? This should be clarified and if data are re-used in different figure this should be clearly stated in legends.

Some data were indeed presented in multiple Figure panels for comparison purpose. This was the case for control conditions when experiments were run in parallel and shared the control (across panels B, D, and F in Figure 1, as well as between Figure 2B and 3D). We have clarified this in the respective Figure legends.

(3) Figure 2 and 3 – Some genetic backgrounds are used that are claimed to change the intracellular Ca++ levels in the FLP neuron with reference to the literature. Since the authors have the capacity to conduct Ca+ imaging in the FLP neuron, using a radiometric indicator, validating the expected effect of unc-68(dom13) (increased) and mec-3p::Calbindin (decreased) on intracellular free Ca++ levels in FLP would strengthen the claims of the authors. In the same vein, the various Ca++ channels mutants that do not affect localization should therefore not affect intracellular free Ca++ levels, which might be surprising. I think the correlation of CMK-1 localization with free Ca++ should be further validated in this way.

These are indeed interesting points/suggestions and we have articulated our answer along three lines:

i) UNC-68 gain-of-function: Regarding *unc-68(dom13)*, the published observations we are referring in the manuscript are results from our group (Marques et al. 2019) where we directly addressed the FLP baseline calcium level in this mutant with a ratiometric indicator and found it was indeed elevated as compared to wild type (see Marques et al. Figure 5). The elevated calcium level in FLP is therefore not inferred from the mutation, but a previously established empirically observation.

ii) Calcium buffering: Regarding *mec-3p::Calbindin* impact on intracellular calcium, this is indeed a very important point in order to interpret its impact on CMK-1 subcellular localization. We therefore conducted additional experiments. First, we conducted the suggested direct calcium measurements. To that end, we generated new transgenic animals co-expressing YC2.3 and Calbindin in FLP and compared them to control animals devoid of Calbindin transgene. We observed that Calbindin expression was linked to decreased YFP/CFP values in animal incubated at 28°C (Figure 2 Supplement 2A). These data are in line with a model in which Calbindin may affect CMK-1 subcellular localization by buffering calcium. Next, because calcium buffering in vivo is a central experiment in our study, we though to expand our work beyond the suggestion by the reviewer. We therefore expressed an alternative calcium buffer, Parvalbumin, in a new set of transgenic animals. Like Calbindin, Parvalbumin was previously used in *C. elegans* to buffer calcium (PMID: 23093425). The reason why we initially focused on Calbindin was that a preliminary assessment comparing a *[mec-3p::Calbindin::GFP]* and a *[mec-3p::Parvalbumin::GFP]* fusion constructs had shown a stronger expression for the Calbindin construct. In the new experiments (Figure 2 Supplement 3), we found that Parvalbumin expression in FLP with a *[mec-3p::Parvalbumin]* transgene was sufficient to significantly reduce CMK-1 nuclear accumulation at 28°C, even though the effect was milder than with Calbindin. More importantly, we expressed, in parallel experiments, a

mutant Parvalbumin unable to bind calcium (K92V, D93A, D95A, K97V and E100V), and found that it had no impact on CMK-1 localization. These results indicate that the impact of the Parvalbumin transgene requires its ability to bind calcium and confirm a model in which the nuclear re-localization of CMK-1 in vivo relies on an unrestrained elevation in free intracellular calcium concentration. The Result section was modified to present these new results.

iii) Impact of mutations in candidate calcium channels. In our previous article (Saro et al.2020), we reported that reduction-of-function mutations in the TRP channels OSM-9;OCR-2, the voltage gated channels UNC-2 and EGL-19, as well as the Ryanodine receptor UNC-68 could all partially impair the response to heat stimuli lasting between 30 s and 5 min. However, this timescale is significantly shorter than the 30-90 min required for temperature elevations to cause CMK-1 nuclear accumulation. The question of whether these calcium channel mutations can impact calcium over longer timescale was still open. We conducted the suggested experiments and recorded YC2.3 YFP/CFP ratio across these different genotypes in animals maintained at 20°C or exposed to 28°C for 60 min, using the same transgenic lines previously used for the work in Saro et al.*,* 2020. Interestingly, we found that, despite the mutations could all impair acute stimulus-evoked response, and unlike *unc-68(dom13)* gain-of-function mutation, the YFP/CFP ratio was not significantly reduced in these different animals at baseline or upon prolonged (1 hour) stimulation with heat (Figure 2 Supplement 2B). These observations are in line with the literature about the robustness of calcium homeostasis and the regulation of resting intracellular calcium level (PMID: 12879862 or PMID: 22453936 for a review), showing that key determinants of the long term free-calcium concentration relate more to an “equilibrium” involving the activity of calcium transporter/exchanger and endogenous calcium buffers, and much less on the activity stimulus-dependent calcium channels, which operate by triggering “disequilibriums” one a shorter timeframe. Collectively, our new data indicate that the candidate calcium channel mutations do not significantly impact calcium level on a timeframe relevant for CMK-1 nuclear entry, which might explain their inability to impair this phenomenon. The Result section was modified to present these new results.

(4) Figure 6H-I and Page 24 lines 570-573: The authors refer to the result that cmk-1 mutants fail to express gcy-8 in the AFD neuron. Both cmk-1 NLS^71-78^ mutants and cmk-1 NES^288-294^ mutants did not increase the expression level of gcy-8 when grown at 25deg. However, these results do not fit the other results presented in the manuscript. In figure 6A-F the AFD neuron is shown to exhibit higher Ca+ levels and nuclear localization of CMK-1, when worms are cultivated at lower temperatures and that mutating the NLS^71-78^ lead to cytoplasmic localization of CMK-1. Figure 6H shows that at lower temperatures where CMK-1 is nuclear, the expression of gcy-8 is reduced. Therefore, one would expect that NES^288-294^ (cmk(syb1375)) mutants that increase nuclear localization of CMK-1 would have lower gcy-8 but the results show the opposite. The authors should discuss a detailed model of cmk-1 dependent gcy-8 expression to address this conflict.

Like this reviewer, we would have anticipated a different result, following a simple model in which calcium levels would be directly translated into gene transcription output, via CMK-1 nuclear entry. However, as correctly emphasized by this reviewer, the *gcy-8* reporter expression level does not follow a one-to-one relationship between the CMK-1 localization pattern that one would assume based on this model. An alternative model could be that *gcy-8* transcription regulation does not simply need CMK-1 to enter the nucleus, but CMK-1 ability to shuttle between the nucleus and the cytoplasm. Because *gcy-8* gene regulation is the only well-characterize transcriptional target known to be regulated by CMK-1, we still think that these data are important to illustrate the consequences that the CMK-1 localization mis-regulation has on gene expression. We therefore decided to keep these data, but expanded the corresponding text in the revised manuscript to clarify that these data suggest a more complex model.

“The growth-temperature effect was abolished in both *cmk-1(syb1435)* and *cmk-1(syb1375)* mutants, leaving expression levels closer to the situation in *cmk-1(wt)* animals grown at 25°C. […] Instead, the ability of CMK-1 to operate an active nucleocytoplasmic shuttling or to be localized in the right compartment at specific times might be essential for *gcy-8* gene transcription regulation.”

(5) Figure 4 in vitro assay of IMA-3 and CMK-1 binding are comparing the effect adding Ca++ plus CaM, but I miss a Ca++ only control.

This is indeed a needed control to properly interpret the requirement for Ca^2+^/CaM. We conducted Calcium only controls and found no significant increase (Table 1 in the revised manuscript).

(6) Pages 9-10 and Figure 2. The authors show that CMK-1 nuclear localization is unaffected in mutants backgrounds of the genes unc-13 and unc-31 required for synaptic transmission and neuropeptide signaling. Indeed, these are classical genetic backgrounds used to test whether an effect is cell autonomous. The authors then claim at several occasions, including in the headline of this chapter that the intra-cellular calcium control of CMK-1 localization in FLP neuron is cell autonomous. This claim is not fully supported by the data since: 1. The double mutant unc-13;unc-31 was not tested. 2. The FLP and neuron make gap junction connections to other neurons that may affect their Ca+ levels and CMK-1 localization, however this possibility was not tested (for example by testing mutants in the innexins expressed by FLP). I don't think that new experiments are necessary here but the authors should accommodate for these possibilities in their conclusions and discussions.

This is a legitimate concern. We have revised the text to remove unsupported claims about the cell-autonomous effect (we notably modified the paragraph sub-title). We explicitly mention that we cannot rule out the redundant function of *unc-13* and *unc-31*, nor the implication of electrical synapses at this stage.

“These results suggest that the mechanisms leading to temperature-dependent CMK-1 nuclear re-localization could primarily depend on cell-autonomous heat-evoked FLP activity. However, we cannot rule out a redundant function of *unc-13* and *unc-31* gene products, nor the implication of electrical synapses.”

(7) Several results described in the work suggest that there is a Ca-dependent but NLS^71-78^ -independent pathway that is involved in the nuclear localization of CMK-1 in FLP and AFD neurons: A. in Figure 3D – the addition of the unc-86(dom13) background seem to increase nuclear localization of NLS^71-78^ mutated CMK-1 (although the effect sizes are small.). Similarly, in Figure 2B, unc-86(dom13) could also increase the nuclear localization of CMK-1 in the Ca+ binding domain mutated CMK(WS305S). B. Figure 6B – the effect of mutating the NLS^71-78^ on CMK-1 localization in AFD is not fully penetrant. This is in contrast with the effect of CMK(WS305S). I suggest the authors to test whether these effects are significant and if so discuss this more clearly.

Thanks for emphasizing these relevant observations suggesting that an additional mechanism might exist in parallel to the NLS^71-78^-dependent mechanism under some circumstances. We have conducted further statistical analyses as suggested.

i) Regarding data in Figure 3D (interaction between *unc-68* mutation and NLS^71-78^ mutations in FLP): Indeed, at 20°C, a Bonferroni post-hoc test indicates a significant effect of the NLS mutations in the *unc-68(dom13)* background (*p<.01*). The effect is significantly smaller than in wild type background, as evidenced by a significant interaction effect in a Two-way ANOVA and the result of other post-hoc tests (New Figure 3 Supplement 1).

ii) Regarding data in Figure 2B (interaction between *unc-68* mutation and W305S mutations in FLP): The effect of the W305S mutation is actually not significant in the *unc-68(dom13)* background (*p*=.25 at 20°C and *p=*.15 at 28°C by Bonferroni post-hoc test).

iii) Regarding data in Figure 6B (partial impact of NLS^71-78^ mutations in AFD): Indeed, the impact of the NLS^71-78^ mutations at 15°C is significantly lower than that of the W305S mutation (*p<.01* by Bonferroni post-hoc tests).

Taken together these data support the existence of a second pathway, independent of NLS^71-78^, which would presumably act downstream of calcium elevation and CaM binding. This second pathway does not seem to be engaged in FLP when calcium is elevated by a 90 min incubation at 28°C (full penetrance of the NLS mutation in FLP, Figure 1F). However, this NLS^71-78^-independent pathway may be engaged as a secondary mechanism, in FLP, when calcium is permanently elevated via the *unc-68(dom13)* mutation and, in AFD, when temperature is maintained at 15°C for an overnight. Further studies will be needed to further delineate the exact nature of NLS^71-78^-independent mechanisms.

We have revised the Results section and expanded the Discussion section to address this aspect in a new paragraph:

“In addition to the NLS^71-78^-dependent mechanism, several lines of evidence suggest the existence of one or more additional mechanisms able to promote CMK-1 nuclear entry or retention. […] Further studies will be needed to delineate the exact nature of NLS^71-78^-independent mechanisms, and notably if they also involve IMA-3-dependent nuclear import and alternative NLS such as NLS^297-308^.”

(8) Figure 5D – negative controls of no ATR or worms without the optogenetic construct are lacking. It is possible the response is a general avoidance to light. The control is mentioned in the methods section but should be shown. It is hard to imagine that there was "no response" as worms always exert spontaneous reversal behaviors.

We apologize for the misleading statement about the absence of response. We intended to mean that, in the absence of ATR, there is no significant blue light-evoked reversal response above the spontaneous reversal rate. We have previously performed a number of control experiments in wild type background in this range of light intensities (Schild et al. 2015). In addition, we are now presenting control experiments conducted in the specific transgenic animals (New Figure 5 Supplement 1). The newly presented control data include -ATR control and red light controls. The red light stimuli were used as a convenient control during manual scoring, in order to monitor the reversal frequency in a time window strictly matching that under the blue light conditions.

(9) Figure 5G and H and Pages 23 and 24 – The authors should cite the work of Satterlee et al., Curr. Biol. (2004) also here when describing the effect of cmk-1 mutants on thermosensation responses and on gcy-8 expression pattern in the AFD neuron, both examined in this work (Table 1 and Figure 4).

Corrected.

(10) Figure 5C – instead of doing the regular thermotaxis experiment they use their own variant on it observing the third quartile of worm distribution" and claiming in the methods section that this is a more robust readout to previous methods. This statement should be validated quantitatively.

Regular thermotaxis assays under conditions where AFD has a major role are reported in Figure 6G. The assay in Figure 5C is not a classical thermotaxis assay, but a specific thermogradient-based assay, which we developed some time ago to focus on Noxious heat avoidance (Glauser et al. 2011). The main points are (i) to use starved worms in which the “classical” AFD-dependent innocuous response is inhibited and (ii) to focus on thermal nociception by making worm navigate in a noxious-temperature range. This assay comes with its own set of limitations (which were partially discussed in the original paper). One limitation which we realized later, after performing many more assays and exploring more genotypes, was that the original scoring metrics (heat avoidance index) was not ideal. This index was relying on a ratio between animal counts in two regions defined with a fixed temperature threshold. This threshold was working well to highlighting differences in wild type and with mutants with strong phenotypes. But there are situations where this index lacks granularity and robustness. This is the case when we look at mutants that produce a relatively mild effect and when the worm distribution is marked by a sharp boundary on the high temperature side of the gradient. In the latter case, small differences in the temperature gradients from assay to assay will cause abrupt change in the index values and increase the variability of the score. This is why we decided to revise the way to score the distribution in this assay by using another distribution index: the third quartile of the distribution. Over the dataset considered in this article, the relative error on the third quartile-based index was 8% on average (range 4-14%), whereas that obtained on the Heat avoidance index was 22% on average (range: 15-32%). In the revised manuscript, we have extended the justification of this methodological choice in the Method section.

(11) Please also report absolute YFP and CFP fluorescent levels of the ratiometric Ca++ indicator; are expression levels affected in 15deg vs 25deg, which can be often observed ion transgenes?

Systematic Cameleon expression difference between conditions could indeed bias the analysis. We used extrachromosomal array-containing transgenic animals and paid special attention to pick animals with similar expression levels. Based on our subjective observations, we feel that the fluorescent signals varied much more from animal to animal than across temperature conditions. As quantitative control, we confirmed that the Cameleon expression was not significantly different between recording sets made at 15 and 25°C. To get a metrics reflecting the Cameleon expression level, we summed the CFP and YFP emission signals (after excitation at 405 nm), representing the total fluorescence independently of varying FRET levels. This metrics was not significantly different between the sets of traces recorded at 15°C and 25°C, respectively (15°C: average=517, sem=93; 25°C, average=552, sem=80; arbitrary units; *p=*.73 by Student’s *T*-test). We have clarified the worm picking procedure and this verification in the Methods section.

Reviewer #2:I am happy to support its publication in eLife. I only have a few comments.(12) Can the authors explain why cmk-1(syb1375) mutants did not effectively spread on the assay plates?

We noticed that cmk-1(syb1375) animals tended to coil producing an increased number of reversals, which reduce their dispersion and ultimately the majority of the animals was concentrated around the assay starting temperature. It was not a major issue for classical thermotaxis assay, because they are symmetrical and a significant fraction of animals nevertheless end up “choosing” one side. It was more of an issue in the noxious heat avoidance assay, where the readout relates to the animal’s ability to disperse unidirectionally. We have modified the text to mention the reason why the animals were not spreading well.

(13) The authors may want to include raw data points in the bar graphs such that readers will get a chance to observe the data distribution patterns and sample sizes (n numbers). This has now become an increasingly common practice. Currently, all bar graphs only show error bars without raw data points or n numbers.

Thank you very much for the suggestion. Indeed, it could be helpful to get more information about the number of observations and their distribution. The facts that the Ns were missing in some of the main Figures was an error. We have now included all the Ns in the respective main Figure panels, directly close to each summary data point. The choice of the bar graph was the result of a careful evaluation of the best quantitative representation of the data. Because we had relatively large samples, displaying raw data was sometimes making the data hard to digest. But on the other hand, we agree that it is still valuable information. As a compromise, we have kept the bar graph in the main text such that readers can readily focus on the average CMK-1 behavior, but we have also generated supplemental figures in which we include a violin plot with jittered datapoints overlaid. We believe this will enable different levels of reading based on the specific interest/focus of the readers.

(14) For FLP CMK-1::GFP translocation data, the authors only showed sample images in one place: Figure 1B. All such data were presented as bar graphs without sample images. Images are easier for readers to comprehend. It is not necessary to show sample images for all bar graphs in the main figures, as this will take up too much space, though the rest may go to supplemental figures.

While we agree with the advantage of a visual communication with micrographs, when trying to implement it, we faced the issue that it complexifies the figure design to a point where it gets more confusing than helpful, especially when we have many conditions. In addition, the point of quantifying the signal is really to get a quantitative picture of CMK-1 localization, and shades of grey or green are known to be one of the least appropriate channels to communicate quantitative data. Therefore, we decided not to include sample pictures each time. We also improved the quality of the sample picture reported in Figure 1B (see response to reviewer 1 above).

(15) typos: "addressed" (line 418) and "serves" (line 625).

Corrected.

Reviewer #3:The manuscript is well written and follows a clear experimental logic with well-designed and supportive experiments and data presentation. The data support well the conclusions and claims. On the other hand, the current methodological description and presentation of statistical analysis needs to be improved.(16) Detailed statistical information is missing; I assume the Authors used ANOVA for multiple comparisons (not mentioned in text or figure legends) with post-hoc t-test using Bonferroni correction for multiple testing (mentioned as Bonferroni contrast) so it would be good to include a detailed description of the statistical analysis used, and report the ANOVA result, dF, and post-hoc test results with exact p-values for the data analyzed.

Thank you for spotting these limitations in our report of the statistics. We indeed carried out ANOVAs and posthoc as this reviewer described. We have clarified the tests used in the Method section and reported the requested indexes as part of the new supplementary figures.

(17) the title "Ca^2+^/CaM binding promotes CMK-1nuclear expression…" in not correct, nuclear localization would be more appropriate.

Corrected.